# Metal Coordination Effects on the Photophysics of Dipyrrinato Photosensitizers

**DOI:** 10.3390/molecules27206967

**Published:** 2022-10-17

**Authors:** Paula C. P. Teeuwen, Zoi Melissari, Mathias O. Senge, René M. Williams

**Affiliations:** 1Molecular Photonics Group, Van ‘t Hoff Institute for Molecular Sciences, University of Amsterdam, P.O. Box 94157, 1090 GD Amsterdam, The Netherlands; 2Medicinal Chemistry, Trinity Translational Medicine Institute, Trinity Centre for Health Sciences, Trinity College Dublin, The University of Dublin St James’s Hospital, D08 RX0X Dublin, Ireland; 3Institute for Advanced Study (TUM-IAS), Technical University of Munich, Lichtenberg-Str. 2a, 85748 Garching, Germany

**Keywords:** photochemistry, photophysics, coordination chemistry, metal atom effect, photodynamic therapy, triplet photosensitizer, dipyrrinato complexes, singlet oxygen generation, triplet-triplet annihilation, heavy atom effect

## Abstract

Within this work, we review the metal coordination effect on the photophysics of metal dipyrrinato complexes. Dipyrrinato complexes are promising candidates in the search for alternative transition metal photosensitizers for application in photodynamic therapy (PDT). These complexes can be activated by irradiation with light of a specific wavelength, after which, cytotoxic reactive oxygen species (ROS) are generated. The metal coordination allows for the use of the heavy atom effect, which can enhance the triplet generation necessary for generation of ROS. Additionally, the flexibility of these complexes for metal ions, substitutions and ligands allows the possibility to tune their photophysical properties. A general overview of the mechanism of photodynamic therapy and the properties of the triplet photosensitizers is given, followed by further details of dipyrrinato complexes described in the literature that show relevance as photosensitizers for PDT. In particular, the photophysical properties of Re(I), Ru(II), Rh(III), Ir(III), Zn(II), Pd(II), Pt(II), Ni(II), Cu(II), Ga(III), In(III) and Al(III) dipyrrinato complexes are discussed. The potential for future development in the field of (dipyrrinato)metal complexes is addressed, and several new research topics are suggested throughout this work. We propose that significant advances could be made for heteroleptic bis(dipyrrinato)zinc(II) and homoleptic bis(dipyrrinato)palladium(II) complexes and their application as photosensitizers for PDT.

## 1. Introduction

Metal-ligand complexes are of crucial relevance in nature with practical applications in various fields such as materials science, catalysis, and medicine [1]. Metallodrugs have been used as chemotherapeutic agents, contrast agents, imaging agents or antibacterial agents [2]. A variety of metal complexes bearing either identical or different ligands are widely investigated. Tuning the photophysical, photochemical, photobiological or electronic properties of these complexes can be achieved by ligand functionalization. A notable family of ligands that has gained much attention over the past years is represented by the dipyrrinato moiety [3,4,5,6,7,8]. Dipyrrinato ligands, obtained from dipyrrins or dipyrromethene derivatives (**DPM**), are small organic anions consisting of two pyrrole moieties, linked by a methine bridge and usually occur in a planar configuration (see Figure 1). **DPM** derivatives are π-conjugated systems that can absorb light (450–550 nm) and undergo ^1^(π–π*) transitions. Historically, researchers were interested in dipyrrinato chemistry due to their relevance in pyrrole chemistry and the synthesis of porphyrins [9,10]. The similarity of dipyrrinato ligands with porphyrins is apparent as their conjugated bis(pyrrolic) moiety is the half of the porphyrin macrocycle. Dipyrrinato moieties are well-known for their ability as ligands in coordination chemistry, as they can coordinate with various metal ions and main-group elements upon deprotonation [4,5,6,7]. Dipyrrinato complexes have attracted research interest due to the wide variety of possible structures and means to tune the properties. For example, the luminescence features upon excitation depend greatly on the environment polarity and structure. The **DPM** structure (Figure 1) can be functionalized on the α-, β- and meso-positions. Depending on the metal employed, dipyrrinato complexes can bear one or more ligands and are classified as homoleptic when the ligands are identical or as heteroleptic when ligands are different [4,5,6,7]. Figure 1 shows homoleptic bis(dipyrrinato)- and tris(dipyrrinato) metal complexes, respectively. The boron complexes of dipyrrinato ligands, e.g., the parent 4,4-difluoro-4-bora-3a,4a-diaza-s-indacene (BODIPY, **B-1**, Figure 1), are by far the most widely studied complexes due to their high fluorescence quantum yields and stability in biological environments [11,12,13,14].

Dipyrrinato complexes have unique properties that lend themselves to potential application in different fields. One example is the use of such complexes as catalysts due to their potential to be functionalized and their ability to catalyze reactions [4]. Similarly, they can be used as fluorescent probes and chemical sensors due to their unique photophysical properties [7]. Other potential applications are as laser dyes, in photovoltaics, as luminescent probes, or in bio-imaging and cancer therapy [5,6]. Progress has been made in tuning the photophysical properties of dipyrrinato complexes based on metals other than boron for their use as fluorescent probes [15]. However, the practical applications of (dipyrrinato)metal complexes is still limited, mainly due to the absence of a detailed understanding of their photochemical and photophysical properties [6]. One promising application that has gained attention over the years is their use as photosensitizers (PSs) in photodynamic therapy (PDT) [15,16,17,18]. PDT is a subcategory of phototherapy which involves the activation of a drug (the PS) with light which subsequently reacts with oxygen in the micro-environment resulting in diseased-tissue damage or cell death [19,20,21,22]. This application will be used in this review as a framework to illustrate the effect of the metal on the photophysical properties of (dipyrrinato)metal complexes and how this can be used to improve translational uses. Other considerations may be necessary for different applications of dipyrrinato complexes. Herein the photophysical properties and the application as PSs for PDT of several d-block and p-block dipyrrinato complexes will be discussed. Figure 2 shows the metal (and non-metal) ions covered in this treatise (colored).

## 2. Photodynamic Therapy

Photodynamic therapy (PDT) is a therapeutic method which involves a less invasive means to treat cancer cells. The treatment involves the excitation of a photosensitizer with visible light of a specific wavelength, followed by triplet generation and the formation of reactive singlet oxygen. In PDT, a non-toxic photosensitizer (PS) is injected into the body and selectively accumulates in the targeted tissue. The PS, which is a light-sensitive dye, can be excited by local irradiation of the targeted tissue at a specific wavelength. The excited PS induces a series of photochemical reactions, which leads to specific apoptotic or necrotic cell death of the malignant cells [19,20,21,22]. PDT is a promising technique compared to other treatments, due to its dual selectivity. An ideal PS selectively accumulates in diseased tissue and can be selectively activated by irradiation with light of a specific wavelength, allowing for non-invasive treatments. Advantages of PDT compared to surgery or radiotherapy are the reduced long-term morbidity, cost-effective treatment, short treatment time, and little or no scarring after healing. The drugs used in PDT are triplet PSs, which can efficiently form the triplet excited state (upon excitation) and then act as catalysts in photochemical reactions. In PDT this is mainly a reaction of the PS with molecular oxygen (Figure 3). Besides the treatment of different types of cancer, PDT has also found application in photodynamic antimicrobial chemotherapy (PACT). This technique allows for the treatment of various infections by bacteria, fungi, viruses and parasites [23,24]. PACT has also been shown to work against drug-resistant strains. In general, triplet PSs are also used in photocatalytic organic reactions, photoinduced hydrogen production from water, luminescent oxygen sensing, and TTA-PUC (triplet-triplet annihilation photochemical up-conversion) [25,26].

### 2.1. Mechanism

Towards the use of conventional PDT in a clinical setting, a standard dose of a drug (the PS), a specific dose from a light source and a certain drug-light interval is required, in the presence of oxygen. A light-activated PS can damage its surroundings via two pathways, called Type I and Type II reactions (see Figure 3) [27]. For both reactions the PS is excited from the ground state (S_0_) to an excited singlet state (e.g., S_2_). Next, the compound can undergo intersystem crossing (ISC) to rapidly form a triplet excited state (T_1_), which is a long-lived state. ISC is the process where the PS undergoes a non-radiative transition from the singlet excited state (S_1_) into a triplet state (S_1_ *→* T_1_ or S_1_ *→* T_n_ *→* T_1_) [20]. Two types of reactions may occur from the triplet state; in Type I reaction the PS from the triplet state transfers an electron or proton directly to biomolecules (lipids, proteins, nucleic acids, etc.) via a radical mechanism. The free radicals and radical ions that are generated in this mechanism interact with oxygen (O_2_), which results in the generation of reactive oxygen species (ROS). ROS, such as hydrogen peroxide (H_2_O_2_), the superoxide radical anion (O_2_^−•^) and the hydroxyl radical (OH•), are unstable entities that cause damage to molecules in the cell and can eventually cause cell death and/or induce immune responses or promote anti-angiogenesis. The Type II reactions are based on singlet oxygen generation via a triplet-triplet annihilation (TTA) mechanism. In a Type II reaction the PS in its triplet excited state reacts with the triplet state (ground state) of oxygen (^3^O_2_, ^3^Σ*_g_*), where highly reactive and cytotoxic singlet oxygen (^1^O_2_, ^1^∆*_g_*) is produced which eventually causes cell death [19]. Both other ROS and ^1^O_2_ are highly reactive and have a short lifetime. For example, the lifetime of singlet oxygen in a ’viable, metabolically-functioning and H_2_O-containing cell’ is approximately 3 µs [28]. This emphasizes that both mechanisms result in a highly localized effect. The balance between the two reaction types depends on the PS, the oxygen concentration, the environment, and the affinity of the PS with the substrate. Conventional PDT, via (either Type I or) Type II mechanisms, require the presence of molecular oxygen. A third oxygen-independent mechanism (Type III) has recently been reported [29]. Overall, Type II is considered as the principal mechanism of PDT.

### 2.2. Triplet Photosensitizers

The current clinically approved PSs are mainly cyclic tetrapyrroles, i.e., porphyrins, chlorins, bacteriochlorins, and phthalocyanines [30,31,32]. An example of a porphyrin drug is *Photofrin* (or porfimer sodium, Figure 4) [33]. This drug is one of the earliest clinical PDT agents and was first approved against bladder cancer in Canada. *Photofrin* is still the most studied PS. Most PSs currently used for anti-cancer PDT still have several drawbacks, such as difficult preparation, purification, and modification of these compounds; poor water solubility; aggregation; dark phototoxicity; photobleaching; slow clearance from the body, photosensitivity for the patients and pain. Therefore, it is important to develop alternative PSs. Among others, the use of transition metal complexes with non-tetrapyrrolic ligands is gaining attention [34,35,36]. For example, the Ru(II) polypyridine complex *TLD-1433* (Figure 4) entered phase II clinical trials against non-muscle invasive bladder cancer (ClinicalTrials.gov Identifier: NCT03053635) [37,38]. It is currently a promising non-tetrapyrrolic TM complex tested in an on-going trial and it exemplifies the potential of these TM complexes as PSs for PDT. Furthermore, extensive research publications and literature review articles cover BODIPYs and their derivatives as potential PSs [13,14]. Clearly, it is necessary to understand how exchanging the boron with different metals can be used to enhance and tune the properties of such complexes. Most of the PS development effort focused on increasing the absorption of the irradiated (visible) light, enhancing ISC, enhancing T_1_ generation, and stabilizing T_1_. Several characterization techniques can be used to determine whether a proposed structure has these desired properties. Photophysical properties such as quantum yields, rate constants and lifetimes can be determined by spectroscopic techniques, i.e., steady state absorption and fluorescence spectroscopy, transient absorption spectroscopy (TA), or theoretical methods such as Density Functional Theory (DFT) and Time-Dependent DFT (TD-DFT). One of the main goals in PS development is to obtain a high quantum yield for singlet oxygen after PS excitation, which is the main component that damages diseased tissues. Considering the singlet oxygen formation pathway, several parameters can influence the ^1^O_2_ generation. First, the light must reach the PS which is accumulated at the tumor site and thus the PS should preferably be excited by light of a wavelength in the so-called ’therapeutic window’ (600–800 nm) in order to achieve sufficient tissue penetration. This window is based on the range of wavelengths where water and tissue chromophores such as hemoglobin and melanin do not absorb strongly [20]. The upper limit of the therapeutic window is related to the minimal energy required for singlet oxygen production. Moreover, the shorter the wavelength that is used, the less light can penetrate tissues. That is the reason why absorption of longer wavelengths, or near-infrared (NIR), is desired [39]. *Photofrin*, for example, has an absorption maximum of 630 nm, which corresponds to a penetration depth of a few millimeters. This makes *Photofrin* only suitable for superficial tumors or those that can be reached via endoscopic or fiber optic procedures. Secondly, a sufficient number of PS molecules must be excited and have a good ability to absorb light; therefore, the molar absorption coefficient (ɛ) has to be sufficiently high in order to enter the pathway. A strong absorbance at the particular wavelength is sought to promote enough molecules to the excited state to either enter type I or type II pathways. *Photofrin* has a low molar absorption coefficient (ɛ *=* 1170 M^−1^ cm^−1^), which makes long irradiation times with a higher energy source necessary and can lead to prolonged skin photosensitivity after treatment (4–12 weeks). Related to this, low (but non-zero) quantum yields for fluorescence are preferred. Fluorescence causes depletion of the singlet excited state and photobleaching leads to inactivation of the chromophore, both resulting in a decreasing number of singlets ready to be converted into triplets via ISC. The fluorescence quantum yield should, however, still be high enough for use in diagnostics and imaging. Going to the next step, a high singlet-to-triplet ISC efficiency is desired to generate enough molecules in the triplet state. Next, the triplet excited state lifetime should be sufficiently long-lived to allow reactive interactions. Finally, a subsequent triplet state energy transfer is needed to excite molecular oxygen from its ground state (^3^Σ*_g_*) to the excited one (^1^∆*_g_*). Therefore, the energy of the triplet state of the PS should be higher than that of singlet oxygen (0.98 eV [20]) to efficiently produce moderate singlet oxygen yields. All parameters and mechanisms mentioned can be tuned to influence the generation of singlet oxygen. The efficiency of a PS does not only depend on the photophysical properties mentioned before, but also on the physicochemical and biological properties, i.e., photostability, (photo)cytotoxicity, hydrophobicity, etc. [13]. For example, a high light/dark cytotoxicity ratio is necessary to allow enough damage in the presence of light and to prevent damage to cells in the absence of light. Besides that, a high tumor cell specificity, to both capture and retain the PS, is desired to let the PS accumulate in the targeted tissue and allow for selective damage; thus, an appropriate lipophilic/hydrophilic balance is necessary [40]. This is important for efficient uptake of the PS into the tissues, good bioavailability and easy administration of the PS [41]. These properties can be optimized by modifying the structures of potential PSs, by adding functional side groups or using a drug delivery system, e.g., incorporating into nanoparticles [14,41,42,43]. Additionally, a low production cost and easy administration of the drug and irradiation are important. Other practical aspects that need to be considered are side-effects (such as skin photosensitivity and pain during and after irradiation), dark toxicity and metabolism of the PS as well as the chemistry of the PSs [20,41,44,45].

## 3. The Heavy Atom Effect in BODIPYs and Chlorins

In this section we clarify the heavy atom effect, by using BODIPYs and chlorins as examples. A significant body of work is available on the use BODIPYs as triplet photosensitizers and their use as promising alternatives to the currently clinically accepted photosensitizers [13,14]. BODIPYs are the most popular dipyrrinato complexes currently under scrutiny; however, regular BODIPYs still have some drawbacks such as small Stokes shifts (5–15 nm) and strong fluorescence [46]. A small Stokes shift can cause self-absorption, which can decrease the detection sensitivity [47]. Enhancing the absorption properties of BODIPYs is possible by modifying their core structures. One example is replacing the carbon atom on the 5-position of the dipyrrinato ligand with a nitrogen atom, forming the so called aza-dipyrrinato ligand. The properties of aza-BODIPYs resemble those of regular BODIPYs; however, they generally show more red-shifted (longer) absorption wavelengths than regular BODIPYs (650–675 nm) [48]. This is beneficial for PDT, as absorption at longer wavelengths can result in deeper tissue penetration [49]. Besides that, side groups such as electron withdrawing or donating groups or hydrophilic/hydrophobic chains can be attached to the core structures to (de)stabilize certain MO-levels (the energy of certain molecular orbitals) or to adjust water solubility [14]. An important goal of the structural modification of BODIPYs is to enhance ISC. Unmodified BODIPY-PSs usually have high quantum yields of fluorescence with relatively low yields of ISC and the absorbed light energy is typically released via fluorescence from the singlet states. This results in a decreased triplet state formation via ISC and thus low singlet oxygen quantum yields. In the ISC process the spin of an electron is reversed resulting in two parallel electron spins, a quantum mechanically forbidden transition. Spin–orbital coupling (SOC) is an important interaction which couples the two spin states in a way that the total energy and the total angular momentum are conserved [50]. The SOC is related to the ISC rate and thus the *T*_1_ generation. Equation (1) displays the estimation of the ISC rate constant ($k_{ISC}$):(1)$$k_{ISC}\propto \frac{<T_{1}\left|H_{SO}\right|S_{1}{>}^{2}}{{\left(\Delta E_{S_{1}-T_{1}}\right)}^{2}}$$
where $H_{SO}$ is the spin–orbit Hamiltonian; $\Delta E_{S_{1}-T_{1}}$ is the S_1_−T_1_ energy gap; $\langle T_{1}\left|H_{SO}\right|S_{1}\rangle$ is the spin–orbit matrix element between the initial (S_1_) and the final (T_1_) wave function. This term depends on the nature (such as shape) of the orbitals involved in the transition. The El Sayed rule in photophysics indicates that ISC is most efficient if the transition involves a change in molecular orbital character (e.g., n *→ π*) [51]. The SOC term contains orbital torque, which means that large SOC values can be attained when the transition involves a change of orbital orientation, which is generally the case for a change in orbital character [52]. The equation also indicates that $\Delta E_{S_{1}-T_{1}}$ is crucial and that ISC takes place efficiently between singlet and triplet states with a small energy gap. This can be referred to as the “energy gap law” [53]. This law states that the transition probability depends on the energy gap. A larger energy gap corresponds with smaller overlap between the vibrational wavefunctions. This results in a smaller transition probability and thus less efficient ISC. SOC is a relativistic effect and arises from the interaction between spin magnetic moment of an electron and the magnetic field from the motion of the nucleus. The nuclear magnetic field depends on the nuclear charge and thus SOC is directly proportional to the atomic number (∝ Z^4^). Following equation 1, one approach to enhance SOC (and thus ISC) in triplet PSs is the insertion of a heavy atom, with a large Z such as halogens or metals, to the chromophore core. This is the so-called heavy atom effect (HAE): *Enhancement of the rate of a spin-forbidden process by the presence of an atom of high-atomic number that is either part of, or external to, the excited molecular entity* [54]. Except for the ISC enhancement, introducing heavy atoms can induce a shift of the PS absorption to longer wavelengths. Longer absorption wavelengths are beneficial for PDT as it allows for deeper tissue penetration [39]. Generally, the SOC values are dependent on many different factors, such as the Z-value, changes in molecular orbital character, and electron distribution. This makes the determination of the exact nature of SOC challenging. An example of such a case is shown in Figure 5 where two BODIPY structures are displayed. The influence of the heavy atoms can be investigated by comparing the BODIPY structure **B-2** with its analogue **B-3**, which has iodo substituents on the β-positions. It was found that **B-3** has a significantly lower fluorescence quantum yield ($\varphi_{f}$) and a higher relative ^1^O_2_ generation rate, which are desired for PSs [55]. It was also observed that adding iodine atoms resulted in a red-shifted absorbance, which is beneficial for PDT applications. In another study it was found that attaching iodine to the 3 and/or 5 positions would induce steric strain and thereby interfere with the planarity of the system, which counteracts the benefits of the heavy atom effect [56]. It should be mentioned that a drawback of attaching iodine is that it could lead to an elevated dark toxicity [57,58]. Incorporation of transition metal ions can be used also to enhance photophysical properties [59,60,61,62]. Coordination with a transition metal ion can create an internal heavy atom effect, also resulting in efficient ISC. An example is the Zn(II)-chlorin shown in Figure 6. Chlorins are tetrapyrrolic structures, similar to porphyrins, except that one of the pyrrole rings is reduced. It was shown that the Zn(II) chlorin **Chlorin-2** displayed an increased ISC rate, a higher triplet excited state quantum yields and a more efficient ^1^O_2_ generation than its free-base counterpart **Chlorin-1** as a result of the internal HAE [63].

## 4. Transition Metal Triplet Photosensitizers

Progress has been made over the years on investigating TM-complexes as triplet PSs alternatively to currently accepted PSs, for applications such as PDT [34,35,36]. It was shown that the introduction of metal ions into clinically accepted PSs could sometimes overcome the present limitations and drawbacks [64,65]. For example, the palladium containing bacteriochlorins *Tookad* and *Tookad Soluble* have improved properties for PDT compared to their free base analogues due to the HAE, such as enhanced ISC, red-shifted absorption and enhanced singlet oxygen generation [66]. Therefore, it is worth studying other metal-complexes as potential alternative PSs. Besides the ISC enhancement, the possibility of tuning the properties by using different metals with the same ligands or modifying the core structure (e.g., by adding side groups or different ligands) makes these compounds very interesting. Lastly, the metal coordination can lead to other excited state electronic configurations and new (possibly more efficient) pathways for triplet generation in addition to those present in organic PSs. These configurations can be metal-centered (MC), within a single ligand (intra-ligand, IL) or involve a charge transfer (CT) state between them. Such CTs can be metal-to-ligand (MLCT), ligand-to-metal (LMCT), within a ligand (ILCT) or between two different ligands (LLCT) [37]. The majority of complexes studied as PSs for PDT are based on Ru(II), Os(II), Ir(III), Rh(III) and Re(I) [15]. Researchers in this field are trying to expand the variety of dipyrrinato complexes in order to find alternatives with cheaper, more abundant and biologically relevant metal ions, for example Fe(II), Zn(II), Cu(II) and Ni(II). Several characterization techniques can be used to study the photophysical and photochemical properties of triplet PSs, and the influence of metal coordination. In the next section some of these techniques will be discussed briefly.

### 4.1. Characterization Techniques for Triplet Photosensitizers

#### 4.1.1. Spectroscopic Studies

Several spectroscopic techniques can be used to characterize triplet PSs. UV-Visible absorption spectra are used to study the ground-state properties of the PS, and the influence of introducing metals in these compounds. Studying absorption spectra of triplet PSs is important for applications such as PDT, as a red-shifted absorption profile can result in stronger absorption of light at longer wavelengths and thus the possibility of deeper tissue penetration. Dipyrrinato complexes are known to absorb in the visible region of the electromagnetic spectrum, due to the *π*-conjugation in these bis-pyrrolic systems [6]. Upon excitation of dipyrrinato complexes, the dipyrrinato ligand undergoes a low energy ligand centered *π-π** transition (S_0_ → S_1_). For nonfunctionalized simple coordinated dipyrrinato ligands the absorption band can be found in the region of 450–500 nm [67]. In bis(dipyrrinato) complexes this would result in the formation of a [D*-Zn-D] state, which can undergo different types of photochemical processes such as charge transfer. In both bis-and tris(dipyrrinato) complexes exciton coupling of the *π-π** transitions can occur, leading to two non-degenerate excited states [68,69,70]. This can be observed in the UV-Visible absorption spectra of these complexes as splitting of the absorption bands (Davydov splitting). Such features are not observed in complexes with a single chromophore [69]. Circular dichroism (CD) spectroscopy is another means to determine whether exciton coupling occurs. Fluorescence spectroscopy investigates the emission profiles and the fluorescence quantum yields and lifetimes can be determined. For PDT low fluorescence quantum yields are desired, as discussed previously. Besides that, decreased fluorescence lifetimes indicate faster depletion of the excited states towards triplet states, which is also desired. The fluorescence quantum yield ($\varphi_{f}$) is the ratio between the number of photons emitted and the number of photons absorbed and can be obtained from the fluorescence spectrum by comparing it to a reference fluorophore with known quantum yield [71]. It can be calculated using the equations below:(2)$$\varphi_{f}=\varphi_{f,ref}\cdot \frac{F_{ref}}{F}\cdot \frac{I}{I_{ref}}\cdot {\left(\frac{n}{n_{ref}}\right)}^{2}\;$$
(3)$$F=1-10^{-A}\;$$
where F is the fraction of light absorbed; I is the integrated emission intensity; n is the refractive index, and A is the absorbance at the irradiation wavelength. The subscript *ref* refers to the reference fluorophore. Examples of a reference with a known $\varphi_{f}$ are quinine sulfate ($\varphi_{f}=$ 0.55 in 0.5M H_2_SO_4_), fluorescein ($\varphi_{f}=$ 0.925 in 0.1M NaOH), and rhodamine 6G ($\varphi_{f}=$0.95 in ethanol) [71]. The fluorescence rate constant ($k_{f}$) can be obtained by
(4)$$k_{f}=\frac{\varphi_{f}}{\tau_{S}}$$
where $\varphi_{f}$ is the fluorescence quantum yield and $\tau_{S}$ is the singlet state lifetime. $\tau_{S}\;$is related to the fluorescence decay and can be measured by time-correlated single photon counting (TC-SPC). Another important parameter is the Stokes shift, which is the difference between positions of maxima of the absorption and emission spectra (resp. λ*_a_* and λ*_f_*, both in nm). It can be calculated in two ways. Either via the difference in maximum wavelengths (Equation (5)) or via the difference in wavenumbers (Equation (6)) given by
(5)$$\Delta \lambda ={\;\lambda }_{a}-\lambda_{f}$$
(6)$$\Delta ṽ={ṽ}_{a}-{ṽ}_{f}$$

Equation (6) is preferred over Equation (5), since the Stokes shift given in wavelength units is dependent on the position of the absorption peak. It is thus less meaningful to give the Stokes shift in nm as comparing Stokes shifts of different compounds is difficult in that case. The correct way to obtain the λ*_a_* and λ*_f_* in cm^−1^ is by converting the complete spectrum from nm to cm^−1^. An easier and quicker way is the use of the following equation:(7)$$\Delta ṽ=\left(\frac{1}{\lambda_{a}}-\frac{1}{\lambda_{f}}\right)\cdot 10^{7}$$

This equation does not take into account a correction needed to account for non-linearity when converting the spectra from a wavelength scale to an energy scale. This can result in a deviation from the correct value of several cm^−1^ [72]. Equation (7) is, however, still useful where no automatic program is available to convert the spectra or when the corrected Stokes shift is not reported. A large Stokes shift is related to less spectral overlap between absorption and emission spectra, which is desired for PDT. More overlap between these spectra results in a reduced fluorescence emission intensity due to self-absorption, which decreases detection sensitivity [73]. Even though the depletion of the singlet excited state via fluorescence is undesired for the PDT mechanism, there is still some fluorescence needed for imaging. Studying triplet excited states is important to assess the efficiency of the triplet PS. One technique that is widely used to study these is the nanosecond time-resolved transient absorption spectroscopy (ns-TA) [74]. With this technique, the triplet state quantum yield ($\varphi_{T}$) can be calculated, which is directly related to the efficiency of ISC ($\varphi_{T}=\varphi_{ISC}$). Other variables that can be obtained are the triplet state lifetime ($\tau_{T\;}$); the ISC rate constant ($k_{ISC}$); the IC quantum yield ($\varphi_{IC}$), and the IC rate constant ($k_{IC}$). The PS (through the triplet excited state) reacts with oxygen via TTA, which results in singlet oxygen generation. Here, oxygen acts as a quencher (*Q*) of the PS T_1_ state. The efficiency of TTA is related to this quenching, and can be evaluated by the Stern-Volmer quenching equation [75,76]:(8)$$\frac{\tau_{T,0}}{\tau_{T}}=1+K_{SV}\left[Q\right]=1+K_{SV}\cdot pO_{2}$$
where $\tau_{T,0\;}$is the triplet-state lifetime of the PS in the absence of oxygen; $pO_{2}$ is the partial pressure of oxygen (in mmHg); and $\tau_{T\;}$ is the quenched triplet-state lifetime. $K_{SV}$ is the Stern-Volmer constant and can be defined by $K_{SV}=k_{q}\cdot \tau_{T,0}$, where $k_{q}$ is the rate constant for quenching of the triplet state by oxygen (s^−1^·mmHg^−1^). A larger $K_{SV}$ leads to more efficient quenching. Lastly, the singlet oxygen quantum yield ($\varphi_{\Delta }$) can be calculated from its phosphorescence via Equation (9):(9)φΔ=mol O12 formedmol photons absorbed=φΔ,ref·SSref·I0,ref·FrefI0·F,
where F is the fraction of light absorbed and $I_{0}$ is the light intensity of their radiation source [77]. To obtain S, several singlet oxygen phosphorescence spectra (signal around 1270 nm) are measured for different irradiation intensities. S is the slope of the linear regression of the plot of the areas of these peaks against the irradiation intensity. An example of a reference compound that can be used is [Ru(bipy)_3_]Cl_2_ in D_2_O ($\varphi_{\Delta }$= 0.22, bipy = 2,2′-bipyridine) [78]. An indirect measurement of $\varphi_{\Delta }$ is also possible. In this technique a molecule acts as a trap and captures the ^1^O_2_, then the change in the absorbance spectrum is monitored through time and relatively compared with a reference compound [79]. More techniques are available for investigating the excited states to characterize triplet PSs, such as time-resolved photoacoustic spectroscopy or laser induced optoacoustic spectroscopy [80,81]. Furthermore, for PDT there are several other parameters that are interesting to study which have not been discussed, such as stability in different solvents, (photo)stability, (photo)cytotoxicity, ROS generation and photochemotherapeutic activity. Photostability can be obtained by measuring UV-Vis intensity ($\lambda_{max}$) over time in the dark or under constant irradiation. Photocytotoxicity can be obtained by comparing cell viability at the start and after a certain amount of time in the dark or under constant irradiation. Finally, the ROS generation can be confirmed by techniques such as electron spin resonance spectroscopy (ESR), which uses ROS scavengers to detect the presence of different types of ROS [82]. Photochemotherapeutic activity (such as antitumor or antibacterial activity) can be accessed via cellular assays [18,83,84].

#### 4.1.2. DFT and TD-DFT Calculations

Other parameters cannot be obtained from spectroscopy but can be approximated with theoretical models such as Density Functional Theory (DFT) and time-dependent DFT (TD-DFT). These methods can predict the structural, energetic, and spectroscopic properties of compounds for the ground-state and excited-state properties of PSs. First, the geometry of the relevant compound is optimized, using a certain functional (usually a hybrid functional such as B3LYP or M06 [63,85]) and a basis set. Generally, basis sets leading to more accurate results cost significantly more computational effort. The compounds discussed herein contain at least one heavy atom (the metal ion). These compounds are difficult to compute because of relativistic effects, especially 3d row TM complexes, and therefore require often more specialized basis sets. The basis sets 631(+)G* or cc-pVDZ can be chosen for all non-metal elements, while a more specialized one for the metal-ions is sometimes chosen. An example is using a pseudopotential (also called an effective core potential) such as cc-pVNZ-PP, which can describe the core electrons of heavy elements such as transition metal elements [86,87,88,89,90,91]. HAE is dependent on the singlet–triplet energy gap and on spin–orbit coupling. When the optimized ground state geometry (minimized energy) is obtained, TD-DFT calculations can be performed to calculate the energy levels (and geometries) of the excited singlet and triplet states. From this the singlet–triplet energy gap ($\Delta E_{S_{1}-T_{1}}$) can be computed; the molecular orbitals can be calculated, and their electron-density distribution can be visualized. Next to the S_1_ → T_1_ transition, it is also possible to investigate transitions between higher excited states such as S_1_ → T_2_, S_2_ → T_1_ and S_2_ → T_1_, see Figure 7. However, higher level excited states are often inaccessible with regular spectroscopy due to fast IC decay. Figure 7 shows that six different pathways for T_1_ generation can be drawn from the energy levels. However, the pathway following S_0_ → S_2_ → S_1_ → T_1_ (green) is still the most probable pathway for T_1_ generation according to the El Sayed and Kasha rules [51,85,92]. In addition to MO energies, the electron-density distribution can also be visualized as orbital shapes with DFT. Especially the orbital shapes of frontier orbitals (HOMO-1, HOMO, LUMO and LUMO+1) of both singlet and triplet states (such as S_1_, S_2_, T_1_ and T_2_) are interesting and commonly studied. Lastly, in order to study the HAE, it is important to obtain information on spin–orbit coupling (Equation (1)). SOC values ($\langle \psi_{S}\left|H_{SO}\right|\psi_{T}\rangle$) for all possible ISC transitions can be computed and compared using this technique [85]. It has become clear that it is quite difficult to exactly predict the SOC values. SOC is dependent on multiple different factors besides Z, such as the mean cubic radial distribution of the electron (∝ *r*^−3^), possible charge transfer processes and changes in the nature of the molecular orbitals. In conclusion, DFT and TDDFT are important tools when assessing and elucidating the role of the metal on the photochemistry of potential PSs.

### 4.2. Enhancing TM Complexes as Triplet Photosensitizers

Some problems must still be resolved for TM complexes to reach their full potential as PSs for PDT. The metal coordination often results in a low absorption in the visible range and short-lived triplet excited states. By coordinating different ligands to the metal or adding substituents to the ligands, photophysical properties can be enhanced. In this way the intrinsic properties of the complex, such as the molecular orbital levels or geometry parameters, can be tuned in a way which benefits the application as a triplet PS.

#### 4.2.1. Enhancing the Molar Absorption Coefficient

Even though TM complexes show efficient ISC, the molar absorption coefficient (ɛ) still poses a problem. TM complexes usually show low molar absorption coefficients in the visible range (ɛ < 10.000 M^−1^ cm^−1^). A possible solution is to attach a visible light-harvesting chromophore (for example BODIPY) to the coordination center [25]. This antenna chromophore is chosen for its strong absorption of visible light and its ability to efficiently funnel the harvested energy to the coordination center to generate triplet states. A method has been reported which allow for the preparation of transition metal complexes that show strong absorption of visible light where a visible light-harvesting antenna is attached to the coordination center [25,93]. This creates a new level, called the ligand-localized singlet excited state (^1^IL). This state is further removed from the coordination center and therefore displays a reduced internal HAE, where instead of the coordination center, now the antenna absorbs light from a specific wavelength. The new ^1^IL state should be higher in energy than the singlet excited state of the coordination center (^1^MLCT) to allow for energy transfer (via internal conversion (IC)) from the antenna to the coordination center (^1^IL → ^1^MLCT) (Figure 8A). This is an allowed transition and thus the IC process is very efficient. Similarly, the ISC of ^1^MLCT → ^3^MLCT is efficient as the coordination center still displays the HAE. Because of the reduced HAE for the ligand centered states, the ISC of ^1^IL → ^3^IL is not efficient. In this way the antenna allows for a strong absorption in the visible range resulting in a high triplet excited state quantum yield. The condition mentioned before, E(^1^IL) > E(^1^MLCT), is somewhat of a limitation. This could be eliminated by attaching the antenna via a *π*-conjugating bond, which could lower the absorption wavelength and thus increase the ^1^IL energy level. It should be taken into consideration that due to the addition of extra conjugation the ligand could become too bulky and as a result the ISC could become less efficient.

#### 4.2.2. Enhancing Triplet Excited State Lifetime

A long triplet lifetime and the presence of oxygen is needed to generate sufficient ^1^O_2_ via TTA. Long triplet lifetimes are especially important in tumor tissues, which usually have a hypoxic environment (lower O_2_ concentration) due to a more active metabolism [95]. Unfortunately, the triplet excited states of TM complexes are usually short-lived. The reason is that besides an increase in the ISC rate, the HAE on the other hand increases the rate of other radiative and nonradiative transitions which deplete the triplet state population, such as phosphorescence (T_1_ → S_0_). This acceleration of other transitions can result in short-lived triplet states. Multiple strategies have been developed to access long-lived triplet excited states of transition metal complexes. Different methods can be used and adjusted to obtain long-lived triplet states [25]. To obtain long-lived triplet states, the phosphorescence should be significantly decreased. One approach is to significantly decrease the energy level of the ^3^IL state. When the ^3^MLCT and the ^3^IL energy states are far apart (such as in Figure 8A), a slight lowering of the ^3^IL state does not influence the phosphorescence lifetime ($\tau_{p\;}$) and QY ($\varphi_{p}$) significantly. Instead, the energy level of the ^3^IL state should be sufficiently lowered until it becomes lower than the energy level of the ^3^MLCT state [94]. In that case, the transition ^3^MLCT → ^3^IL becomes very efficient (Figure 8B) and the phosphorescence from the ^3^MLCT state will be greatly lowered, or even quenched in accordance with Kasha’s rule [92]. The ^3^IL state can then instead of the ^3^MLCTstate react further with oxygen to form singlet oxygen. Phosphorescence of the ^3^IL state is not efficient due to the decreased HAE on the ligand, which results in a long triplet excited state. However, there is a downside to this process; the upward energy transfer via ISC of ^1^IL to ^1^MLCT is very unlikely to occur. Besides that, the ISC between ^1^IL and ^3^IL is not efficient due to the decreased HAE on the ligand. This means that the coordination center should be excited instead of the antenna, which removes the advantage of a strong visible light absorption. Another option is to establish an equilibrium between ^3^IL and ^3^MLCT. It is envisaged that if the energy level of the ^3^IL state is close to that of the ^3^MLCT state an equilibrium between the two will be established and this will extend the triplet state lifetime. With that approach the advantage of strong absorption in the visible range is not lost and the phosphorescence is still reduced. Several complexes, with different metals, ligands, and antennas, have been investigated in this context [25,95]. However, here the theory will be applied in the context of dipyrrinato complexes.

#### 4.2.3. Practical Example: Ru(II) Photosensitizer *TLD-1433*

Efforts have been focused on the development of transition metal complexes as potential PSs for use in PDT. The most studied TM complexes for this purpose are based on Pt(IV), Ru(II), and Rh(III), followed more recently by Ir(III) and Os(II) [37,65,96,97,98,99,100,101,102,103,104,105,106]. The potential of TM complexes as PSs, Ru(II) complexes in particular, is supported by PS *TLD-1433* which has entered human clinical trials (ClinicalTrials.gov Identifier: NCT03053635) [37,38]. *TLD-1433* was introduced as a therapeutic agent for non-muscle invasive bladder cancer and was first administered to a patient in March 2017 in Toronto. It was shown that *TLD-1433* was almost 200 times more selective for tumor cells than for healthy tissue. The light-source was placed inside the bladder using an optical fiber. From the development process of this compound valuable considerations can be obtained which may be beneficial in developing new TM dipyrrinato complexes as PSs for PDT [37].*TLD-1433* is a chloride salt of a racemic Ru(II) complex with one *α*-terthienyl (3T) substituted imidazo [4,5-f]-1,10-phenanthroline (IP) ligand and two 4,4-dimethyl-2,2-bipyridine (4,4′-dmb) ligands [37]. Multiple considerations were relevant in the choice of this PS out of a selection of related complexes. Research started with [Ru(bipy)_3_]^2+^ complexes, mainly because extensive work and information on the photophysical properties and synthetic procedures were available from the use of this compound in photovoltaics and catalysis [103,107]. By *π*-expansion of one of the bipy-ligands of [Ru(bipy)_3_]^2+^, it was shown that a ^3^IL state could be introduced. The researchers who developed *TLD-1433* studied the sensitivity of this ^3^IL state to oxygen. As discussed in the last section, the energy of the ^3^IL state should be lower than the ^3^MLCT state to lead to an accessible state with a long triplet lifetime. It was determined that the type of conjugation mattered for the energy of this ^3^IL state. Without *π*-expansion, such as regular [Ru(bipy)_3_]^2+^, a situation similar to that shown in Figure 8A is present. Introducing *π*-expansion along the M-N coordinate resulted in a low-energy ^3^IL state, like the situation in Figure 8B. An energy below 2.1 eV could be obtained, which corresponded to their goal. Another condition was a triplet lifetime of >20 µs, which was also possible by this structural modification. Introducing *π*-expansion perpendicular to the M-N axis did not result in the anticipated improvements and even lowered the ^3^MLCT state, which resulted in a decreased lifetime. Different types of conjugations were trialed, for example, fusing the functional ligand with more aromatic rings or tethering an organic ligand with or without a linker. Choosing two or three functional ligands was also possible, but by selecting one functional ligand aqueous solubility could be maximized and aggregation could be reduced (even though using more ligands would lead to longer lifetimes). The conjugated substituent that was chosen was *α*-terthienyl, resulting in the structure for *TLD-1433*. Figure 9 shows the energy diagram for *TLD-1433* together with the transitions involved. This figure also shows the transition to the less sensitive ^3^MLCT state (which is emissive), which is beneficial for imaging purposes. It also shows the TTA process between low-lying ^3^IL state and oxygen for the generation of singlet oxygen. The choice of the structure, type of substituents, conjugation, the substitution on the ligands (e.g., methyl substituents), the type of functional ligand (IP), and counter-ions (e.g., Cl^−^) were based on aspects such as maximizing aqueous solubility, reducing aggregation, biological compatibility or tunability. Besides these pharmacological and chemical reasons, a crucial benefit was cost reduction. Cost could be reduced by choosing the structure with a lower molecular weight and a simpler or higher yielding synthetic procedure. Besides that, a structure needs to have a unique composition and utility in order to obtain a patent on it. These considerations are crucial in determining whether a compound can have potential as a new medicine. This highlights the importance of investigating new alternative transition metal complexes with different types of ligands (such as dipyrrinato ligands).

## 5. Dipyrrinato Complexes as PSs for PDT

Dipyrrinato ligands are known to form stable complexes with a variety of metal ions, from all over the periodic table (s/p/d block). Multiple review articles have been published about dipyrrinato complexes, their synthesis, and luminescence properties [3,4,5,6,7,8]. In the following sections some of the d- and p-block TM dipyrrinato complexes and their triplet-chemistry will be discussed, apart from boron dipyrrinato dyes (BODIPY) which are well-known and extensively investigated. In Section 5.1, d-block dipyrrinato complexes will be described. In Section 5.2, p-block dipyrrinato complexes such as Ga(III), In(III) and Al(III) dipyrrinato complexes will be described. Dipyrrinato complexes with s-block metals such as calcium, magnesium, lithium, sodium and potassium have been previously reported; however, to date they do not display any capability as PSs and limited information can be found with regards to their geometries and optical properties [7,108,109,110,111,112]. Lithium dipyrrinato salts could, however, be used as precursors in the synthesis of other dipyrrinato complexes [6,113]. Several complexes have been synthesized with dipyrrin-derived ligands such as. N_2_O_2_- and aza-dipyrrinato (Figure 10). N_2_O_2_-type ligands are similar to regular dipyrrinato ligands but with phenol groups attached on the pyrrolic *α*-positions, allowing for tetradentate binding with the central metal atom. N_2_O_2_-type ligands have been shown to form complexes with d-block TMs such as Zr, Ti, Mn, Co, Ni, Cu, Pt, and Au, etc. and with the p-block elements B, In, Ga, Si, Ge, Sn, and Al [6,114,115,116,117,118,119]. Aza-dipyrrinato ligands consist of a nitrogen atom instead of the carbon in the methine bridge. They have been shown to form complexes with Cu(I), Ag(I), Au(I), Zn(II), Re(I), Co(II), Ni(II), Hg(II), Rh(I), Ir(I), Ir(III), Li, Na, K, B, and P [6,120,121,122,123]. Aza-BODIPYs display a red-shifted (NIR) absorption and fluorescence profile (*λ_a_ ≈* 650 nm, λ_e_
*≈* 675 nm) with a lower fluorescence quantum yield compared to regular BODIPYs, which may be beneficial for PDT [13]. Bis- and tris(dipyrrinato) complexes are excellent materials for the formation of supramolecular and polymeric architectures. This is due to the presence of coordination bonds that can lead to self-assemblies of organic ligands and metal ions (or clusters) into crystalline network structures. Several frameworks, for instance coordination polymers, macrocycles, and metal organic frameworks (MOFs), based on dipyrrinato-complexes as functional units have been investigated [8,10,124,125,126]. MOFs have been investigated as drug delivery systems in PDT due to its porous network that can be used to encapsulate molecules minimizing the aggregation and enhancing singlet oxygen yields. Another advantage is that the physicochemical properties of MOFs can be modified, such that they can present good biocompatibility and can be degraded in the organisms. Finally, loaded MOFs can enhance the solubility of PSs and increase cellular uptake [127].

### 5.1. d-Block Dipyrrinato Metal Complexes

Dipyrrinato ligands have shown the ability to form complexes with multiple late TMs (group 8–11); for example, homoleptic complexes with Ni(II), Pd(II), Fe(II), Fe(III), Co(II), Co(III), Cu(I), Cu(II), Rh(III) and Ir(III) have been investigated [5,6,8,15,18,67,83,128,129]. TMs from group 12, such as Zn(II), are ’post transition metals’ and have also been shown to form dipyrrinato complexes. Besides that, several luminescent heteroleptic dipyrrinato complexes have been investigated containing Cu(I), Re(I), Pt(II), Pd(II), Ir(III), Rh(III), Zn(II) and Ru(II) [6,7,16,17,67,130]. Dipyrrinato complexes have been investigated mostly for their geometry and their luminescence properties (UV-Vis absorption spectra and emission spectra). Besides Re(I), few dipyrrinato complexes with other early TM’s (group 1–7) have been investigated. For dipyrrinato complexes with Mn(II), Mn(III), Mo(VI) and Cr(III) only few photophysical data have been reported, apart from the occasional UV-Visible absorption spectra [108,131,132,133,134]. In the next sections various d-block metal dipyrrinato complexes with potential applicability towards photomedicine will be discussed. In particular, the photophysical properties of Re(I), Ru(II), Rh(III), Ir(III), Zn(II), Ni(II), Cu(II), Pd(II) and Pt(II) dipyrrinato complexes will be described.

#### 5.1.1. Re(I), Ru(II), Rh(III) and Ir(III) Dipyrrinato Complexes

Rhenium (Re)-based complexes with ligands other than dipyrrinato ligands, e.g., pyridine, polypyridyl, tricarbonyl, pyridocarbazole or phenanthroline ligands, are among the most studied TM complexes as PSs for PDT [135,136,137,138,139] Most of the complexes show enhancement of ISC and oxygen quantum yields due to long ^3^MLCT state lifetimes. Their potential mechanisms of PDT include: (i) phototoxicity; (ii) DNA binding; (iii) enzyme inhibition; (iv) mitochondrial effects or (v) oxidative stress regulation. Many ligands have been described that be used to tune the lipophilicity, the luminescent properties, the cellular uptake, the biodistribution, the cytotoxicity, the pharmacological and toxicological profile. Nevertheless, only few articles have been published based on the photophysical properties of dipyrrinato)Re(I) complexes (Figure 11) [16,130,140,141].

Telfer and co-workers investigated a range of Re(I) complexes with dipyrrinato ligands and different numbers of CO- or phosphine ligands (**Re-1** and **Re-2**, Figure 11) [16]. These complexes showed strong absorption in the visible range (470–480 nm, ɛ up to 4.2 ∗ 10^4^ M^−1^ cm^−1^); however, for each complex a weak emission from the *π-π** band ($\varphi_{f}$ = 0–0.006 at *≈* 700 nm) and large Stokes shifts (*≈* 6000 cm^−1^) were observed. No exciton coupling was observed, as these were mono(dipyrrinato) complexes. It was shown that exchanging PPh_3_ phosphine-ligands with PBu_3_ ligands caused a blue-shift of the *π−π** band in the UV-Vis absorption spectrum. Additionally, by increasing the number of phosphine ligands from one to two resulted in a blue-shift and a decrease in the absorption intensity. Formation of a ^3^MLCT state and phosphorescence from a dipyrrinato-centered triplet excited state was suggested by DFT computations. However, Raman spectroscopy did not provide strong evidence for this. This phosphorescence was shown to be sensitive to oxygen; however, triplet lifetimes were not reported. Following these results, the same group synthesized more (dipyrrinato)Re(I) complexes with several meso-substituents (**Re-3**) [140]. None of the complexes discussed in this article showed any detectable luminescence. It was suggested that this could be due to photo-induced intramolecular charge transfer processes; however, this was not investigated further. **Re-1** and **Re-2** were found to be emissive; this is perhaps due to the complexation with the pyridine. The article did not state in which solvent the emission was measured. Recently, Manav et al. investigated the singlet oxygen efficiency and photostability of a range of Re(I) complexes (**Re-4**, Figure 11) [130]. These complexes showed strong absorption in the UV-Visible range, very weak fluorescence with large Stokes shift values (5600–6960 cm^−1^). In addition, these complexes displayed long triplet lifetimes ($\tau_{T\;}$ = 9–29 µs) and high singlet oxygen yields ($\varphi_{\Delta }$ = 0.75–0.99). The highest singlet oxygen yield was obtained for the complex with a *p*-fluorophenyl group (**Re-4b,**
$\varphi_{\Delta }$ = 0.99), with N-butylcarbazole in the second place (**Re-4g,**
$\varphi_{\Delta }$ = 0.98). Additionally, it was found that the electron donating or withdrawing ability of the substituents influenced the structural, electrochemical, and spectroscopic properties of these complexes. The substitution at the meso position did not influence the position of the absorption maxima of this band; however, the substituents did have an influence on the phosphorescence wavelengths. To illustrate, compounds with bulky electron rich aromatic groups (**Re-4f,g,h**, Figure 11) showed phosphorescence between 681 and 692 nm with relatively smaller Stokes shifts. In contrast, electron withdrawing halobenzene groups resulted in maxima at higher wavelengths around 698–736 nm. The complex with the strongest electron withdrawing group, pentafluorophenyl (**Re-4a**), exhibited the most red-shifted absorption profile. In conclusion, their long triplet state lifetimes, efficient singlet oxygen generation and distinct photo-stability makes them good candidates as potential PSs for PDT. Complexes based on bivalent ruthenium ions (Ru(II)) have been extensively studied for different applications due to their relative stability and their unique photophysical properties. Notably, the properties and applications of [Ru(bipy)_3_]^2+^ have been well studied [103,107]. Much research has been dedicated to the application of Ru(II) complexes in PDT, such as *TLD-1433* discussed previously [37,96]. Most of these compounds have good water-solubility, long luminescence decays, high singlet oxygen yields, and high chemical and photophysical stability [37]. An example of Ru(II) complexes used for PDT are blue-green absorbing Ru(II) complexes with π-conjugated ligands as reported by Yin et al. [142]. These complexes exhibited long-lived triplet lifetimes and they could be activated with red/NIR light to yield PSs for multiwavelength PDT. Remarkably, these complexes showed photodynamic efficacy when they were excited at longer wavelength (625 nm), regardless of their low absorptivity at this region (ε < 100 M^−1^ cm^−1^). Although these complexes absorb outside the photo therapeutic window (< 500 nm), they displayed promising potential as PSs for PDT, with light EC50 values of 0.4–1.9 μM against HL60 human promyelocytic leukemia cells. Another article described a red-shift in the absorption of Ru(II) polypyridyl complexes towards the therapeutic window. This was achieved via the introduction of suitable π-conjugated moieties on the [Ru(bipy)3]^2+^ core, such as extension with methyl groups or vinyl dimethylamino groups, presenting phototoxicity against cervical cancerous HeLa cells [143].

With regards to (dipyrrinato)Ru(II) complexes, several heteroleptic complexes have been studied [6,17,67,144,145,146,147,148,149,150]. The focus of these articles was on their synthesis, photophysical properties, antitumor activity, and applications in dye-sensitized solar cells. The heteroleptic Ru(II) dipyrrinato and bipyridine complexes with carboxyl, carboxylate, or ester as functional groups (**Ru-1a-d** and **Ru-2a,b** in Figure 12) have been investigated [67,145,146]. **Ru-2a** displayed two distinct bands in the visible region (Figure 13) [67,145]. The sharp, intense band at 454 ([67]) or 480 nm ([145]) was assigned to the intra-ligand dipyrrinato (*π−π**) transition. The second, broader band at 635–638 nm, was ascribed to a Ru(II) → bipy (MLCT) transition. This Ru(II) → bipy band was red-shifted compared to the corresponding MLCT transition of [Ru(bipy)_3_]^2+^ (443 nm) [151]. Similar bands were observed for **Ru-1d** (first peak at 483 nm and second at 512 nm); however, for this compound the red-shift was less prominent compared to **Ru-2a**. Therefore, this red-shift could be explained by raised d-orbital energies due to the anionic and weak *π*-accepting character of the dipyrrinato ligands (more dipyrrinato ligands = more red-shift). Exciton coupling was not (clearly) visible in the absorption spectra of both **Ru-1d** and **Ru-2a**. Complexes **Ru-1a-d** were weakly emissive and Raman spectroscopy indicated that only weak electronic interactions were present between the MLCT and *π−π** transitions (even though they were both coordinated to the Ru(II) metal ion). Excitation of the MLCT bands of **Ru-2a** and **Ru-2b** did not lead to any emission, which is suggested to be caused by decay via low-lying ligand field states. In addition, complexes **Ru-3** have been developed without further characterization apart from X-ray analysis [144]. In the same study, the homoleptic tris(dipyrrinato)Rh(III) complexes **Rh-1a** and **Rh-1b** were investigated and their properties were compared to **Ru-1d** and **Ru-2a** [67]. Using XRD, **Rh-1a** and **Rh-1b** were found to adopt either a distorted or pseudo-octahedral geometry (respectively) with approximately D_3_ symmetry. **Rh-1a** displayed two intense absorption peaks at 460 and 498 nm, corresponding to the dipyrrinato *π−π** transition (Figure 13). The splitting of this band was ascribed to strong exciton coupling due to the presence of three dipyrrinato ligands in close proximity [68,69,70]. In contrast to the non-emissive (dipyrrinato)Ru(II) complexes, **Rh-1a** and **Rh-1b** were found to be luminescent upon excitation at 458 nm (although with a small quantum yield) with large Stokes shifts (*λ_f_* = 544 nm, ∆ṽ = 3450 cm^−1^) [67]. The emission profiles of both complexes were broad (spanning >100 nm) and relatively weak. The difference between the luminescence properties of (dipyrrinato)Ru(II) and Rh(III) is the accessibility of the low-lying MC states. Due to the larger ligand field splitting of Rh(III) these states can become inaccessible, thus removing some non-radiative deactivation pathways. The tris(dipyrrinato)Co(III) complex **Co-1** was found to be virtually isomorphous to its analogue **Rh-1a** and was determined to be non-emissive [128,152]. Exciton coupling effects, similar to those observed for **Rh-1a**, were also observed for **Co-1** [152].

Swavey et al. reported monometallic (**Ru-4)** and trimetallic (**Ru-5**) (dipyrrinato)Ru(II) complexes with extended dipyrrinato ligands as possible PSs using the human lung cancer cell line A549 [17]. Both complexes showed absorption at ~ 290 nm that belong to π-π* transition of bipyridine moieties. Complex **Ru-5** showed higher intensity due to the increased number of bipyridines. Complex **Ru-4** displayed a band at 570 nm with a shoulder at 540 nm due to an overlap of transitions by the dipyrrinato ligands and a transition overlap of the ruthenium and bipyridines MLCT transitions, respectively. Complex **Ru-5** had two distinct bands at 502 and 578 nm that presumably belong to the ruthenium to bipyridine MLCT transition and the dipyrrin intra-ligand charge transition, respectively. The latter can also include an overlap of dipyrrin ILCT and Ru(II) dipyrrin MLCT transitions. In addition, **Ru-5** had another absorption band at 350 nm, most likely connected with a higher energy MLCT transition originating from the peripheral Ru(II) centers. Both complexes could generate singlet oxygen sufficiently to operate via type II mechanism by irradiation within the PDT window or at higher energy (420 nm). The in vitro (photo)cytotoxicity was evaluated and both complexes did not exhibit any dark toxicity against lung cancer cells (A549 cell line) up to 50 μM. Upon irradiation of the cells (~ 420 nm, 2.3 μW cm^−2^) **Ru-4** did not show any phototoxicity; however, complex **Ru-5** displayed significant phototoxicity at 50 μM concentration. This was explained by the fact that **Ru-5** generated 50% higher light-induced ROS compared to the control. Interestingly, photocleavage studies with supercoiled plasmid DNA (pUC18) demonstrated that the complexes initiated DNA photodegradation (irradiation > 550 nm). It was reported that **Ru-5** reacted with DNA under hypoxic conditions causing DNA photodamage, which is crucial in PDT since in cancer cells and tumorous sites oxygen concentrations are low [17]. Furthermore, heteroleptic Ru(II) dipyrrinato complexes with (methoxypyridyl)phenyl, methylthiophenyl, pyrimidyl-piperazine, *p*-cymene and ferrocene substitution have been reported to demonstrate binding affinity to DNA or proteins and act as anticancer agents [147,148,149,153,154,155]. For example, Paitandi et al. prepared and investigated several mono(dipyrrinato)Ru(II), -Rh(III) and -Ir(III) complexes with ferrocenyl substituents for use as anticancer agents (**Ru-(6–8), Rh-2** and **Ir-1,**
Figure 14) [149]. UV-Visible spectra of un-complexed dipyrrin ligands displayed two weak absorption bands at 480 nm (Fe−Cp) and at 395 nm due to π–π* charge transfer transitions, respectively. The Ru(II), Rh(III) and Ir(III) complexes showed strong absorbance at 500–510 nm which were assigned to π-π* charge transfer transitions from the dipyrrin ligand. Additional weaker absorbance was observed at 418–450 nm due to the MLCT transitions. Intense high energy transitions at 340–350 nm were assigned to intra-ligand π-π* transitions. Interactions of the complexes with calf thymus DNA (CT-DNA) and BSA (bovine serum albumin) have been assessed via UV–Vis absorption, ethidium bromide displacement fluorescence studies, synchronous, and 3D fluorescence spectroscopy. In addition, molecular docking studies proposed that the complexes bind with the minor groove of DNA and are located within the subdomain IIA cavity of the protein. In vitro anticancer studies showed that the complexes **Ru-(6–8), Rh-2** and **Ir-1** induced cytotoxicity and apoptosis against Dalton’s lymphoma (DL) cell line. **Rh-2** was the most toxic and could induce apoptosis even at low concentrations. The toxicity of these complexes was observed according to the following descending order: **Rh-2**(IC_50_ = 20–30 μg mL^−1^) > **Ru-8** (IC_50_ = 80–90 μg mL^−1^) > **Ru-7** (IC_50_ = 100–110 μg mL^−1^) > **Ru-6** (IC_50_ = 110 μg mL^−1^) > **Ir-1** (IC_50_ = 110 μg mL^−1^). The same group worked on a new set of Ru(II) organometallic complexes **Ru-(9–12)** (Figure 14) [147]. UV-Vis absorption spectra of these complexes displayed three bands: a strong band at low energy at 490–510 nm corresponding to ^1^π-π* charge transfer from the conjugated dipyrrin core; another band at higher energy at 420–460 nm assigned to MLCT transitions; and a third band in UV region at 305–340 nm that was ascribed to dipyrrin-based intra-ligand π-π* transitions. Following, the DNA binding affinity of the demonstrated the interaction of the complexes with CT-DNA through intercalation in the DNA and molecular docking studies suggested that they bind with the minor groove of the DNA same as the previous work. Finally, a significant cytotoxicity against A549 cell line was present for the complexes **Ru-(9–12)**, inducing apoptosis efficiently. The most efficient complex was the pentafluorophenyl derivative **Ru-10**, with the lowest IC_50_ value of 20 μM. Gupta et al. developed four new heteroleptic complexes **Ru-13, Ru-14, Rh-3** and **Ir-2**, with (η^6^-arene)Ru(II)-, (η^5^-C_5_Me_5_)Rh(III)-, and (η^5^-C_5_Me_5_)Ir(III)- moieties and 4-(2-methoxypyridyl)phenyldipyrrinato (Figure 14) [153]. Their absorption spectra displayed an intense low energy band ~ 485–500 nm that was assigned to conjugated dipyrrin ^1^π-π* transition and MLCT transitions, whereas the high energy bands at ~260 nm and at ~350 to dipyrrin-based intra-ligand π-π* transitions. Their DNA binding activity was evident even at very low concentrations. The cytotoxic efficiency against DL cells of these complexes based on binding constant and antitumor activity was as follows: **Ru-14** (IC_50_ = 5–10 μg mL^−1^) > **Ru-13** (IC_50_ = 8–10 μg mL^−1^) > **Ir-2** (IC_50_ = 30–40 μg mL^−1^) > **Rh-3** (IC_50_ = 75–100 μg mL^−1^). The same group evaluated a new class of mono(dipyrrinato) complexes based on Ru(II), Rh(III), and Ir(III), containing 5-(4-methylthiophenyl)dipyrrin (**Ru-15**, **Ru-16**, **Rh-4**, **Ir-3;**
Figure 14) [154]. They exhibited strong low energy absorptions at ~490–500 nm; weak bands at ~385–430 nm in the absorption spectra; and high energy intense bands at ~250 nm which have been ascribed to the intra-ligand ^1^π-π* transitions. Similar to the previous reports these complexes appeared to act as intercalators in the DNA (through the minor DNA groove) as they bind efficiently through intercalative or electrostatic interactions. In vitro anti-cancer activity of the complexes was remarkably improved and they showed higher activation of the endonuclease for DNA cleavage. The descending order was: **Ru-16** (IC_50_ = 5–10 μg mL^−1^) > **Ru-15** (IC_50_ = 8–10 μg mL^−1^) > **Ir-3** (IC_50_ = 30–40 μg mL^−1^) > **Rh-4** (IC_50_ = 75–100 μg mL^−1^). Similar observations were made by this group with arene Ru(II) heteroleptic dipyrrinato complexes containing 5-(2-pyrimidylpiperazine)phenyldipyrrinato and 5-(2-pyridylpiperazine)phenyldipyrrinato, that displayed in vitro cytotoxicity against kidney cancer cell line (ACHN) and suggested an apoptotic mode of cell death [148]. Iridium metal complexes have been widely investigated for applications in catalysis, materials in electronic sensors, photochemistry and luminescent chemosensors or LEDs. The application towards biomedicine is still in its infancy; however, there is an increasing effort in developing Ir(III)-based chelates with biomedical purposes. Most of the Ir(III) complexes related to PDT research are based on polypyridyl ligands. Looking at Ir(III) and dipyrrin ligands only, a limited number of complexes have been investigated in the context of PDT [18,104,105]. Other (dipyrrinato)iridium(III) complexes have been reported in the literature in different contexts [149,150,153,154,156,157,158,159,160]. Particularly, Hohlfeld et al. investigated the application of a wide range of heteroleptic (dipyrrinato)iridium(III) complexes for application in PDT with four tumor cell lines and antibacterial PDT with two bacterial strains known to pose one major problem in hospital infections (Gram-positive germ *S. aureus* and Gram-negative *P. aeruginosa* [18]. In total 30 different (dipyrrinato)iridium(III) complexes were synthesized. These could be grouped into the chlorido(dipyrrinato)(pentamethyl-η^5^-cyclopentadienyl)iridium(III) type **Ir-4,5** or (dipyrrinato)bis(2-phenylpyridyl)iridium(III) type **Ir-6,7** (Figure 15). Both the groups were further divided by two types of aromatic substituents, one with 4R-tetrafluorophenyl (substituent X) and second with 3-nitrophenyl-4R (substituent Y). To investigate the scope of the reaction the complexes were functionalized via nucleophilic substitution, glycosylation, and BODIPY conjugation. Several synthetic procedures were presented to allow easy access to this variety of Ir(III)complexes. Their suitability for PDT was assessed with four cancer cell lines and two bacterial strains. Some of the complexes were found to show high photo-cytotoxicity against tumor cells and strong activity against bacteria, often even without illumination. Overall, it was determined that the complexes **Ir-4,5**, specifically non-functionalized or with simple alkyl chains, proved to be most effective (strong dark and phototoxic effects). Within complexes **Ir-6,7**, the complexes with alkenyl, alkynyl, and polar substituents showed the strongest reduction in tumor cell viability, with saccharide-substituted complexes being most effective. This study illustrated the potential for Ir(III) complexes as PSs for PDT. However, no detailed photophysical studies were reported.

Besides this detailed study on (dipyrrinato)Ir(III) complexes for PDT, no other (dipyrrinato)Ir(III) complexes have been reported in the context of PDT; however, one study focused on the phosphorescence of bis-cyclometalated Ir(III) dipyrrinato complexes with application in OLEDs. A variety of ligands (CN) and substituents on the meso-position of the dipyrrinato ligand were introduced to yield complexes **Ir-(8–11)** [156]. The electrochemical and photophysical properties of the complexes were dominated by the dipyrrinato ligand. The dipyrrinato ligand acted as the chromophore and showed strong absorption of visible light (λ_a_ = 470–485 nm, ɛ *=* 3.8·10^4^ M^−1^ cm^−1^) with large Stokes shifts. The complexes showed phosphorescence at room temperature from a dipyrrin-centered triplet state with quantum yields up to 0.115, triplet state lifetimes of 12.9–23.1 µs and wavelengths from 658 to 685 nm. This red-shifted phosphorescence is positioned in the biological tissue window (same as therapeutic window), which allows for PDT applications. Additionally, it was suggested that the efficient triplet state formation after photoexcitation is caused by the formation of the ^3^IL state [25]. Their high molar absorptivity, large Stokes shift and efficient triplet state formation facilitates possible applications as PSs in PDT. The absorption wavelength (*≈*480 nm) would still be a limitation for PDT, as wavelength falls below the therapeutic window; however, the absorption wavelength could easily be modified by changing the ligands, without affecting the properties of the dipyrrinato ligand. Applications of this or related compounds have not been reported yet.

#### 5.1.2. Dipyrrinato-Zn(II) Complexes

Bis(dipyrrinato)Zn(II) complexes are most studied for their luminescence and photophysical properties. These Zn(II)-based complexes (*d*^10^) usually obtain a tetrahedral geometry. Zn-ions are much cheaper, more abundant, and biologically relevant than most of the TM-ions discussed before, which makes these complexes suitable as alternative PSs for PDT. Several Zn(II) dipyrrinato complexes have been reported (Figure 16, Table A1 and Table A2, see Appendix A) [8,79,110,120,161,162,163,164,165,166,167,168,169,170]. The properties summarized in Table A1 indicate that the bulkiness of the *R*^1^ ligand at the meso-position of the dipyrrinato has a significant influence on the fluorescence quantum yield. This is supported by a much higher $\varphi_{f}$ for **Zn-2g** ($\varphi_{f}$ = 0.36) compared to the phenyl and *tert*-butyl analogues **Zn-2b** and **Zn-2e** ($\varphi_{f}$ = 0.006 and $\varphi_{f}\;$ = 0.007, respectively) [162]. Similar results were obtained for heteroleptic (dipyrrinato)Pd(II) and -Pt(II) complexes [171]. The mesityl side group is torsionally constrained due to steric hindrance from the 2,6-dimethyl groups on the aryl ring. This results in less conformational freedom in the ground or excited states and thus less excited-state relaxations. Because of this a longer excited state lifetime is observed for **Zn-2g** compared to the phenyl-analogue ($\tau_{S\;}$ = 3 ns vs. 0.09 ns). This is also supported by a reduced Stokes shift, a 2-fold greater $k_{f}$ and a 75-fold slower $k_{IC}$ for **Zn-2g** ($k_{f}$ = 0.13 ns^−1^ and $k_{IC}$ = 0.13 ns^−1^) compared to **Zn-2b** ($k_{f}$ = 0.07 ns^−1^ and $k_{IC}$ > 10 ns^−1^). The influence of steric constraints was also observed for the positioning of naphthalene as a side group at either the 1- or 2-position [163]. **Zn-2h** showed 10× stronger luminescence and a sharper photoluminescence spectrum in THF than its analogues **Zn-2b** and **Zn-2i**.

The influence of the substituents at the R^1^, R^2^ and R^3^ positions on the photophysical properties for bis(dipyrrinato)Zn(II) complexes has been investigated. For example, it was shown that the introduction of heavy atom containing side groups such as iodine to the dipyrrinato core-structure can lead to an increased ISC rate and a red-shift of the absorption. Replacing the protons on the R^2^ positions of **Zn-4g** (R^2^ = H) by iodine (R^2^ = I) lead to a shift in λ*_a_* of 419 nm to 517 nm in toluene and resulted in a reduced $\varphi_{f}$ (0.02–0.045 vs. 0.129–0.138 in toluene) and an increased triplet quantum yield (0.63 vs. 0.16 in de-aerated toluene) [79,164,166]. A large $\varphi_{\Delta }$ of 0.61 in toluene was also observed for **Zn-4g** (R^2^ = I). **Zn-5**, a compound with chloro-substituents on the R^1^, R^2^ and R^3^ positions, also showed a high triplet yield of 0.89 in toluene [168]. Concerning the (dipyrrinato)zinc(II) complexes, attention has been risen towards the symmetry breaking charge transfer (SBCT) and the formation of ligand-ligand charge transfer states for complexes in solution (^1^LLCT), which are also called Intramolecular Charge Transfer (ICT) states [164,165,166,167,168]. Triplet generation can be enhanced by tuning this process. For example, ultrafast formation of these ^1^LLCT states is the main reason why **Zn-5** has such a high triplet yield (0.89 in toluene) [168]. The process starts by excitation of the complexes, which results in the formation of a ^1^*π−π** state [D*-Zn-D]. This is followed by intramolecular charge transfer (ICT) from one ligand to another, which breaks the symmetry of the complex by generation of a ligand-ligand CT state (^1^LLCT which looks like ^1^[D^+*•*^-Zn-D^−*•*^] or ^1^[D^−*•*^-Zn-D^+*•*^]. Next, the ^1^LLCT state undergoes charge recombination which generates either the ground state S_1_ (CRS) or a neutral triplet state T_1_ via ISC (CRT). This triplet generation can occur via two mechanisms which are displayed in Figure 17A.The first mechanism, spin–orbit-induced ISC (SOCT-ISC), involves direct ISC from ^1^LLCT to T_1_. The second mechanism, radical pair ISC (RP-ISC), has the same result but involves indirect ISC via a triplet LLCT state (^1^LLCT *→*
^3^LLCT *→* T_1_). In this mechanism the ^1^LLCTand ^3^LLCTstates, are typically almost degenerate and mixed due to electron-nuclear hyperfine coupling and can be determined by investigating the electronic coupling between the donor (*D^+•^*) and the acceptor (*D^−•^*). SOCT-ISC is the preferred mechanism when there is strong electronic coupling. The energy levels of the LLCT states can be altered by structure modifications and in this way SOCT-ISC and the generation of triplets can be tuned. It has been determined that SOCT-ISC becomes more efficient when the distance between the involved orbitals becomes smaller or the orthogonality increases [172,173,174,175]. For applications such as PDT, it is useful to make SOCT-ISC more efficient and in that way to increase the triplet yield. SBCT processes have been investigated for a set of bis(dipyrrinato)zinc(II) complexes with mesityl groups at the meso-position [164]. The mesityl groups are torsionally constrained, which helps in increasing the singlet lifetime and is beneficial for investigating SBCT processes. With this set of bis(dipyrrinato)zinc(II) complexes it was reported that besides the influence of the meso-substitution, also the substitution of the *α-* and β-positions can influence the photophysical properties. For $k_{\mathrm{nr}}$ the descending order was:**Zn-3g** ∼ **Zn-2g** < **Zn-4g** (R^2^ = C_2_H_5_) ∼ **Zn-4g** (R^2^ = H).

Methylation at the R^1^ and R^3^ positions of complexes with Ar = Mes (**Zn-4g**, R^2^ = H, C_2_H_5_) resulted in a significant increase in $k_{\mathrm{nr}}$. It was found that steric interactions between the mesityl side group and these methyl substituents caused an out-of-plane distortion of the dipyrrinato core, which was not seen for **Zn-2g** and **Zn-3g**. It is suggested that this distortion could be an explanation for the differences in $k_{\mathrm{nr}}$. Methylation of the *α*-positions (= R^1^) and ethyl substitution at the β-positions (= R^2^) only caused a marginal decrease in $k_{\mathrm{nr}}$. The presence of ethyl residues at the β positions did, however, result in a red-shift of the absorption with λ_a_ = 506–508 nm for **Zn-4g** (R^2^ = C_2_H_5_) in cyclohexane, toluene and DCM and λ_a_ = 488–495 nm for the other three complexes (**Zn-2g**, **Zn-3g** and **Zn-4g** (R^2^ = H)). This red-shift was also observed upon the attachment of a *p*-tolylacetylide group (TA) at the β-position of a similar Zn(II)-dipyrrin, **Zn-4f** (R^2^ = TA), with λ_a_ = 553 nm in toluene (see Figure 18a) [165]. This is a relatively large red-shift in contrast with the complex **Zn-4f** (R^2^ = I), where HAE occurs due to the iodine substituents; and has a λ_a_ of 516 nm in toluene [79].

It was also reported that solvent has a significant impact on the photophysical properties. Generally lower fluorescence quantum yields and rate constants were observed for compounds in DCM than in toluene. A few of the compounds have also been measured in other solvents with varying polarity [164]. For example, **Zn-3g** exhibited high to moderate yield in nonpolar solvents (0.66 in cyclohexane and 0.19 in toluene) and low yield in polar solvents (<0.01 in DCM). A similar trend was observed for **Zn-2g** and **Zn-4g** (R^2^ = H, C_2_H_5_). This is due to the formation of the ^1^LLCT state, which is not fluorescent, and is favored in polar solvents. LLCT states are charged and therefore destabilized in solvents with decreased polarity (Figure 17B). It was proposed in the article that in this case the ^1^LLCT state is higher in energy than the S_1_ state, which hinders the generation of the ^1^LLCT state. By increasing the polarity of the solvent, the ^1^LLCT state stabilizes to below the S_1_ state and charge transfer becomes more efficient (Figure 17A). When the polarity is increased further, the LLCT states are more destabilized and eventually become lower in energy than the T_1_ state. Here the LLCT states could behave as an ’energetic trap’. This again results in a reduced $\varphi_{T}$. Another study measured the photophysical properties of some bis(dipyrrinato)Zn(II) complexes in the solid state [176]. Spectroscopic studies and single-crystal X-ray analysis showed that the aggregation states of the complexes influence the emission properties. For instance, the packing structure and alignment of the molecules was observed to influence λ_f_. Karges et al. investigated bis(dipyrrinato)Zn(II) complexes **Zn-4b,f,g,j** (R^2^ = I) for application as PSs in PDT [79]. These four complexes absorbed at 516–517 nm, and the iodine atoms promoted SOC resulting in ISC and ^1^O_2_ generation upon irradiation at 510 nm (20 min, 5.0 J cm^−2^) and at 540 nm (40 min, 9.5 J cm^−2^), which was confirmed by EPR spectroscopy. Phototoxicity studies were performed using human retinal epithelial cells (RPE-1), human cervical carcinoma (HeLa), mouse colon carcinoma (CT-26), and human glioblastoma astrocytoma (U373) cells. The complexes were found to efficiently enter cancer cells within 4 h through passive diffusion and accumulated in the cytoplasm. The four complexes showed no dark toxicity, and they were phototoxic against several cancer cell lines at low micromolar concentration. Additionally, they were found to be active in a HeLa MCTS (multicellular tumor spheroids) 3D tumor model. Anthracene-derivative **Zn-4j** (R^2^ = I) displayed the highest cell uptake efficiency. The complexes **Zn-4f,g,j** (R^2^ = I) showed singlet oxygen quantum yields of 0.43–0.57 in MeOH and 0.02–0.05 in aqueous solution. This agrees with the stabilization of the LLCT states in strongly polar solvents and the formation of an ’energetic trap’ as mentioned before. The complexes **Zn-4f** and **Zn-4g** (R^2^ = I) were found to be the most phototoxic. The reference complex **Zn-4b** (R^2^ = I) showed a significantly smaller singlet oxygen yield (0.06–0.10), smaller luminescence quantum yield and a larger Stokes shift in MeOH, compared to **Zn-4f,g,j** (R^2^ = I).

A disadvantage of the iodo-substituted bis(dipyrrinato)Zn(II) complexes is quenching of the excited states in aqueous environments. To improve water solubility, **Zn-4g** (R^2^ = I), which showed the best photophysical and phototoxic properties, was encapsulated in a polymer matrix (1,2-distearoyl-sn-glycero-3-phosphoethanolamine-N-[methoxy(polyethylene glycol)-2000] ammonium salt (DSPE-PEG2000-OCH_3_) [79]. Additionally, the matrix nanoparticles allowed for improved photophysical properties and selective accumulation in lysosomes, contrary to the complex alone that accumulated in the cytoplasm. Upon light exposure, the nanoparticles caused cell death at low micromolar concentrations in the monolayer cancer cells and HeLa MCTS 3D tumor model, presenting the potential of these complexes in PDT. The triplet lifetimes of the iodo-substituted bis(dipyrrinato)Zn(II) complexes **Zn-4b,f,g,j** (R^2^ = I) were determined to range from 107 to 354 ns (in toluene, not degassed), see Table A2 (Appendix A) [79]. **Zn-5** was also studied for the SBCT process in a different study, and a triplet lifetime of 630 ns was observed (in toluene, cyclohexane and acetonitrile) [168]. Even though it was claimed in this study that the iodine substituted Zn(II) complexes display *’exceptionally long lifetimes’*, these $\tau_{T\;}$ are still relatively short compared to other triplet lifetimes discussed in this work. As a comparison, **Zn-2g** showed a higher triplet lifetime of 50 µs in toluene under ambient conditions. However, this is still insufficient for applications such as PDT. Nevertheless, introduction of a bulky conjugated ligand at the meso-position was effective in increasing the triplet lifetime. The same group reported that the heavy-atom free analogues **Zn-4b,f,g,j** (R^2^ = H) showed potential as imaging probes [170]. These complexes had intense fluorescence in non-polar solvents, the emission was quenched in polar environment, and were localized in HeLa cells cytoplasm by passive diffusion. To overcome the quenching and increase the water-solubility, complex **Zn-4g** (R^2^ = H) was encapsulated in a polymer matrix with biotin group. The nanoparticles maintained their bright fluorescence and similarly with the iodinated analogues they selectively accumulated in the HeLa cancer cells lysosomes and fully penetrated 3D MCTS. The four complexes did not show any dark toxicity and did show phototoxicity upon irradiation at 510 nm (20 min, 5.0 J cm^−2^) and 540 nm (40 min, 9.5 J cm^−2^) against RPE-1 and HeLa cells (IC_50_ > 100 μM), similar to the iodo-substituted analogues. Mahmood et al. studied the series of bis(dipyrrinato)Zn(II) complexes **Zn-4b,i,j,k** (R^2^ = H) and demonstrated that their triplet excited state dynamics could be effectively tuned using structural modifications [167]. The anthracene and pyrene substituted complexes **Zn-4j** (R^2^ = H) and **Zn-4k** (R^2^ = H) were reported to have significantly longer triplet lifetimes (295 µs and 146 µs in DCM, resp.) compared to their phenyl and 2-napthalene substituted analogues **Zn-4b** (R^2^ = H) and **Zn-4i** (R^2^ = H) (4.6 µs and 1.4 µs resp.). This was attributed to differences in the energy gap between CT and T_n_ states, along with differences in the geometry by introducing bulky electron donating aryl moieties at the meso-positions of the complexes. The bulky anthracene and pyrene groups hinder intramolecular rotations, leading to slower fluorescence decay. This is supported by smaller $k_{f}$-values for **Zn-4j,k** (R^2^ = H) (0.004 ns^−1^ in DCM) compared to **Zn-4b,i** (R^2^ = H) (0.01–0.03 ns^−1^). A reduced $k_{f}$ is desired as it results in a more efficient ISC and longer triplet lifetimes, and in this way the ^1^O_2_ production can be tuned. The ^1^O_2_ quantum yields were determined to be (in ascending order, R^2^ = H): **Zn-4i** > **Zn-4b** > **Zn-4k** > **Zn-4j** ($\varphi_{\Delta }$= 0.10, 0.13, 0.35 and 0.71 resp.). The solvent effect was also observed by Mahmood et al. [167]. The triplet lifetimes of the compounds were higher in glycerol triacetate ($\tau_{T,k}$ = 743 µs and $\tau_{T,j}$ = 1244 µs), a slightly viscous solvent, which stabilizes the triplet state, likely due to TTA inhibition. This extension of the triplet lifetime was not observed for **Zn-4b** (R^2^ = H), with only $\tau_{T,b}$ = 6.3 µs in glycerol triacetate. Besides that, ISC for **Zn-4j** (R^2^ = H) was negligible in toluene and efficient in DCM, due to the inhibition of the SOCT-ISC mechanism in non-polar solvents. Lastly, ^1^O_2_ generation of these complexes was also investigated: $\varphi_{\Delta }$ = 0.10, 0.08, 0.62, and 0.28 in DCM for **Zn-4b,i,j,k** (R^2^ = H), respectively. ^1^O_2_ generation was found to be smaller in acetonitrile (more polar than DCM) and slightly larger in chloroform (slightly less polar than DCM) (Figure 17A). No ^1^O_2_ generation was observed in toluene (more non-polar than DCM) for all complexes as expected (Figure 17B). Nishihara and coworkers showed that heteroleptic bis(dipyrrinato)Zn(II) complexes **Zn-(6–8)** differ in their photophysical properties from the homoleptic analogues (see Figure 19) [165,177]. The SBCT process in such complexes is not important, due to the lack of symmetry. The emission spectrum of **Zn-6** showed only one emission band for the TA-substituted dipyrrinato (D2) ligand (λ*_f_* = 578 nm), whereas the absorption spectrum was found to be the sum of the absorption bands of the homoleptic analogues **Zn-3g** and **Zn-4f** (R^2^ = TA) (see Figure 18a) [165]. The single emission band was observed after excitation of D1 (495 nm), but also after selective excitation of the other ligand (D2, 553 nm). This suggested the presence of energy transfer from D1 (green) to D2 (blue), followed by emission from a [D1-Zn-D2*] state. Furthermore, an enhancement of $\varphi_{f}$ for **Zn-6** was observed compared to the homoleptic analogues, for non-polar and even polar solvents ($\varphi_{f}$ = 0.76 in toluene and $\varphi_{f}$ = 0.52 in DCM; $\varphi_{f,Zn-3g}$ = 0.28 in toluene and non-emissive in DCM). It was found that the asymmetry and the ordering of the frontier orbitals caused this effect. The results are summarized in Figure 18b,c. For **Zn-6** the HOMO-1 and LUMO+1 are localized on ligand D1 and the HOMO and LUMO are localized on ligand D2. Both CT processes for complex **Zn-6** require a gain in energy of the electron that is transferred. This is less efficient than for the homoleptic complexes, where electron transfer occurs along degenerate frontier orbitals. This suggests that for heteroleptic complexes the ligand-to-ligand CT is less efficient, even in polar solvents, resulting in a less efficient formation of non-emissive ^1^[D1^+*•*^-Zn-D2^−*•*^] and ^1^[D1^−*•*^-Zn-D2^+*•*^] states and thus an increased $\varphi_{f}$. Another publication elucidated the fluorescence properties of several heteroleptic bis(dipyrrinato)Zn(II) complexes with an even larger range of substituents [177]. The best performing complex was **Zn-7**, which showed a high luminescence quantum yield of 0.62–0.72 in both polar and nonpolar solvents. In addition, a NIR absorption maxima (λ_a_ = 671 nm in toluene), a large ɛ of 89,000 M^−1^ cm^−1^ or 180,000 M^−1^ cm^−1^ (one for each ligand) and large Stokes shifts of *±*5400 cm^−1^ were achieved.

Not all heteroleptic complexes result in higher fluorescence quantum yields compared to their homoleptic analogues. For example, for **Zn-8** an opposite effect was observed [165]. This complex showed a significantly lower $\varphi_{f}$ in non-polar and especially polar solvents ($\varphi_{f}$ = 0.07–0.08 in toluene and $\varphi_{f}$ = 0.01–0.03 in DCM). This was again explained by investigating the orbital energies with DFT, see Figure 18d. The introduction of methyl and ethyl groups, which are electron donating, causes a destabilization of the HOMO and LUMO on the D1 ligand. Consequently, the electron transfer process [D1-Zn-D2*] → ^1^[D1^+*•*^-Zn-D2^−*•*^] occurs by the transfer of an electron from a higher energy level to a lower energy one. This process is more efficient than the processes seen for the other complexes, where the energy levels have the same height (Figure 18c) or the electron has to go from a lower to a higher energy level (Figure 18d). The generation of the non-emissive ^1^[D1^+*•*^-Zn-D2^−*•*^] state is therefore also efficient, which results in a decreased $\varphi_{f}$. The effect of these changes on the triplet generation was not mentioned. It is hypothesized that the incorporation of electron donating groups could lead to an enhancement of triplet generation in heteroleptic bis(dipyrrinato)Zn(II) ligands. The introduction of dissymmetry could therefore be a new efficient tool for enhancing the triplet generation in these types of complexes. The goal of Nishihara and coworkers was to enhance the fluorescence of bis(dipyrrinato)Zn(II) complexes, which was shown to be possible by introducing dissymmetry in the complexes. Following on, they investigated a heteroleptic complex (**Zn-9**), consisting of a regular dipyrrinato ligand and an aza-dipyrrinato ligand coordinated with a Zn(II) metal ion (Figure 20) [120]. Aza-BODIPYs, which contain an aza-dipyrrinato ligand (Figure 10) complexed with a boron, have been shown to have valuable properties compared to their regular dipyrrinato analogues, such as a red-shifted absorption profile and larger Stokes shifts [48,49]. Therefore, using aza-dipyrrinato ligands instead of regular dipyrrinato ligands may also improve the luminescent properties of heteroleptic (dipyrrinato)Zn(II) complexes. **Zn-9** was shown to feature a lower fluorescence quantum yield ($\varphi_{f}$ = 0.00043) compared to the homoleptic analogue **Zn-3a** ($\varphi_{f}$ = 0.054) and higher than the homoleptic analogue **Zn-10** (no detected luminescence). The absorption spectra and emission spectra of **Zn-9**, **Zn-3a**, and **Zn-10** can be found in Figure 20. Similar to **Zn-6**, the absorbance spectrum of **Zn-9** is the sum of those of its homoleptic analogues. After excitation of either of the ligands (490 nm for the dipyrrinato and 600 nm for the aza-dipyrrin), the complex exclusively emits at 672 nm. This indicates energy transfer from the dipyrrinato (D1) to the aza-dipyrrinato (D2). Additionally, the fluorescence weakens in more polar solvents ($\varphi_{f}$ = 0.00043 in DMF and $\varphi_{f}$= 0.0001 in DCM) indicating that a mechanism similar to **Zn-6** (Figure 18c) is involved. These results show that a non-emissive aza-dipyrrinato complex can be made emissive (at long wavelengths) by introducing dissymmetry via replacing one aza-dipyrrinato ligand with a regular dipyrrinato ligand. To conclude, quenched fluorescence is observed for homoleptic bis(dipyrrinato)Zn(II) complexes in polar solvents due to the occurrence of SBCT. This quenching effect can be suppressed and fluorescence can be improved in heteroleptic compared to homoleptic analogues. The effect of combining two different dipyrrinato ligands on triplet formation is yet unknown. However, it was observed that the generation of LLCT states, which are precursors in the mechanisms for triplet formation, is hampered in most heteroleptic complexes that were studied. The introduction of electron-donating groups on one of the dipyrrinato linkers, similar to **Zn-8**, could possibly improve triplet formation; however, this has never been investigated.

#### 5.1.3. (Dipyrrinato)-Pd(II) and -Pt(II) Complexes

Bis(dipyrrinato)-Pd(II) and -Pt(II) complexes are mainly investigated structurally together with their luminescence properties (**Pd-(1–6)**, Figure 21) [67,171,178,179]. Crystal field theory states that *d^8^* complexes (such as Pd(II) and Pt(II) complexes) prefer a square planar configuration due to the maximalization of the crystal field stabilization energy (CFSE) [180,181]. The stabilization obtained by a square planar configuration can be counteracted by destabilization caused by steric interactions between ligands, such as between *α*-substituents in the pyrrolic rings of dipyrrinato ligands. In order to release the strain, the structure can reorient itself and distort away from the square planar configuration in order to allow for more room for the *α*-substituents [182]. According to crystal field theory, the strictness of this preferred geometry depends on the crystal field splitting and thus on the specific metal and ligands. In general, the magnitude of the crystal field splitting tends to increase for increasing principal quantum number (n) (and thus size) [180,181]. For larger metals the steric constraints have a smaller effect as the ligands are further away from each other. Therefore, complexes with Pt(II) metal ions (5*d*^8^) are expected to show a stricter preference for the square planar geometry compared to complexes with the smaller Pd(II) metal ions (4*d*^8^).

Hall et al. showed that **Pd-6** maintained a strictly square planar configuration (Figure 22) [67]. The strain caused by the inter-ligand contacts of the α-protons was accommodated by forcing the ligands to cant away from the PdN_4_ plane, causing the two dipyrrinato ligands to lose coplanarity. Additionally, a nonplanar distortion of the bis-pyrrolic core was observed to further accommodate strain. **Pd-5** and **Pd-6** displayed intense bands in their UV-Vis spectra around 480 nm, which were ascribed to ^1^π-π* transitions of the dipyrrinato moieties (Figure 13). Hall et al. noted that this showed that the curvature found in the dipyrrinato ligands did not significantly perturb their electronic transitions. Exciton coupling was not visible in their absorption spectra due to a null net transition dipole moment for the transition to the lower energy excited state [68,69,70]. The dipyrrinato ^1^π-π* transitions were found to be non-emissive upon irradiation. Recently, Mathew et al. reported the synthesis of **Pd-(1–4)** via a one-pot reaction and a preliminary investigation on their antitumor activity [178]. For these complexes a loss of coplanarity between the dipyrrinato ligands (similar to **Pd-6**) and a curvature of the bis-pyrrolic core were observed. Additionally, the complexes were shown to adopt a slightly additional distorted square planar geometry. Some experimental results are shown in Table A3 (Appendix A). The absorption spectra of these complexes displayed one intense band for *π−π** transitions around 470–490 nm (ɛ = 55,000 M^−1^ cm^−1^). Additionally, a broad band around 350–400 nm (ɛ = 10,000 M^−1^ cm^−1^) was observed, corresponding to a MLCT transition. These complexes displayed both weak luminescence and fluorescence ($\varphi_{f}$ = 0.008–0.031 in acetonitrile, chloroform, and toluene). Their weak fluorescence is supported by short singlet state lifetimes (2.5–3.8 ns). Triplet state lifetimes were obtained with transient absorption spectroscopy for **Pd-(1–4)** (45 ns, 29 ns, 32 ns, and 68 ns, respectively). The average singlet oxygen quantum yield in all the solvents was relatively low ($\varphi_{\Delta }\;\approx 0.08$), with a highest value of $\varphi_{\Delta }$ = 0.170 found for the phenyl derivative **Pd-1** in acetonitrile. Additionally, their triplet excited state lifetimes were also observed to be short (30–70 ns). It was suggested that the out-of-plane distortion of the Pd(II) complexes leads to low-lying, metal-centered (MC) d-d excited states which provide efficient non-radiative deactivation pathways.

In the same study, in vitro studies of **Pd-(1–4)** showed cytotoxicity against DLA cancer cell lines [178]. For all four complexes, no noticeable impact on non-cancerous rat spleen cell lines was observed, which indicates a selectivity for the action of these complexes. The cyanophenyl **Pd-3** was found to be the most cytotoxic (IC_50_ value of 50.73 μg mL^−1^), while complex **Pd-4** was found to be the least cytotoxic (IC_50_ value of 72.01 μg mL^−1^). The authors suggested that the anticancer activity did not stem from their photosensitizing ability but due to the nuclear DNA binding ability, similar to cisplatin (also a Pd(II) complex). These complexes have not been investigated yet in the context of PDT; however, with further development they could reveal potential as an anticancer agent. Another report by Riese et al. on complex **Pd-1** showed that this complex undergoes a major structural reorganization [179]. After excitation, the complex undergoes a reorganization from a square planar geometry to yield a disphenoidal (seesaw) triplet state, which was supported by DFT calculations. Both fluorescence decay studies and DFT calculations showed the involvement of ^1^LC_2_ and ^1^LC_1_ (ligand-centered) excited states where after excitation and the population of ^1^LC_2_ a short-lived singlet state is created which quickly relaxed to ^1^LC_1_. In this state a partially allowed fluorescence was observed, likely due to the structural flexibility of **Pd-1** that disturbed the symmetry. The triplet state was dominated by a fast ISC process from the LC states with an ISC rate constants of ca. (13–16 ps)^−1^. Lastly, the absence of any phosphorescence emission of **Pd-1** was validated by the structural reorganization leading to a non-emissive triplet metal centered state (^3^MC). A comparison can be made between the photophysical properties of **Pd-1** (in MeCN) and its direct Zn(II) analogue **Zn-2b** (in toluene) [161,162,178]. **Pd-1** displayed a slightly smaller absorption wavelength (477 nm vs. 485 nm), a larger molar absorption coefficient (1.78 *×* 10^5^ M^−1^ cm^−1^ vs. 1.15 × 10^5^ M^−1^ cm^−1^), similar fluorescence quantum yield (0.008 vs. 0.006), and a larger Stokes shift (2565 cm^−1^ vs. 660 cm^−1^) (Table A1 and Table A3, Appendix A). Interestingly, the singlet oxygen quantum yield of **Pd-1** (0.17 in MeCN) is slightly larger than both Zn(II) complexes **Zn-4b** (R^2^ = H) and **Zn-4b** (R^2^ = I) (0.10 in DCM and 0.10 in MeCN, respectively) [79,167,178]. These properties, especially the higher quantum yield without iodo-substituents, indicate that similar bis(dipyrrinato)Pd(II) complexes may also show promise for application in PDT. It is suggested that triplet state life times and singlet oxygen quantum yields of **Pd-1** can be enhanced in similar ways as described for bis(dipyrrinato)Zn(II) complexes. For example, the substitution of iodine at the R^2^ position and/or the substitution of mesityl, pyrene or anthracene residues at the R^1^ (meso) position could be attempted [79,167]. The effect of such modifications on the photophysical properties of bis(dipyrrinato)Pd(II) complexes has never been investigated before. Bronner et al. synthesized and studied the photophysical properties of four new luminescent heteroleptic (dipyrrinato)Pd(II) and -Pt(II) complexes that combine a mesityl or benzonitrile dipyrrinato ligand with a cyclometalated 2-phenylpyridine (ppy) ligand (**Pd-7,8** and **Pt-1,2**, Figure 21) [171]. X-ray crystallography showed that geometry of the Pd(II) center of **Pd-7** and **Pd-8** deviated slightly from square planar. Each pyrrole ring of the dipyrrinato ligand was determined to lie either above or below the plane defined by the Pd(II) and the 2-phenylpyridine (ppy) ligand. A deviation from planarity in the bis-pyrrolic core of the dipyrrinato ligands was observed, similar to **Pd-(1–6)**. The geometry around the Pt(II) center of **Pt-1** and **Pt-2** was strictly square planar. As a result, the strain induced distortion and a loss of coplanarity between the two pyrrolic rings. This observed distortion was smaller for **Pt-2** than for **Pt-1**. Interestingly, the beforementioned observations indicate that the Pd(II) complex **Pd-8** was not isostructural to its Pt(II) analogue **Pt-2**. This supports the earlier notion based on crystal field theory that Pt(II) complexes are expected to have stricter preference for the square planar conformation compared to Pd(II) complexes. The four complexes (**Pd-7,8** and **Pt-1,2**) showed a strong absorption band around 500 nm, corresponding to the *π−π** transition of the dipyrrinato ligand. A second band around 430 nm was only evident for the Pt(II) complexes which was attributed to the lower oxidation potential of Pt(II) compared to Pd(II), which enables a transition to a MLCT state. Nevertheless, their luminescence was determined to be weak, even at lower temperatures as low as 77 K. Amongst the complexes, the Pt(II) analogues exhibited the strongest fluorescence emission. This indicates that the singlet excited states of Pt(II) complexes undergo faster radiative processes and slower non-radiative deactivation processes. To support this, the excited state lifetimes of the Pt(II) complexes were determined to be two orders of magnitude higher than for the Pd(II) complexes. This may be due to the out-of-plane distortion featured by the Pd(II) complexes that can result in low-lying MC d-d excited states which favors certain deactivation pathways, similar to what was suggested for **Pd-(1–4) [178]**. The strongest fluorescence emission was observed for mesityl derivative **Pt-1**, which is related to the restriction of rotational freedom of the mesityl substituent as previously described for Zn(II) bis(dipyrrinato) complexes [162].

#### 5.1.4. Bis(dipyrrinato)Ni(II) and -Cu(II) Complexes

Complexes based on the bivalent metal ions nickel (Ni(II)) and copper (Cu(II)) generally favor square planar or tetrahedral configurations, depending on the nature of their ligands [169]. Steric repulsion from *α*-substituents can be accommodated by increasing the inter-ligand dihedral angles and thereby creating intermediate geometries between square planar and tetrahedral [15,169,184,185,186,187,188,189,190,191,192,193,194]. According to the Brunings-Corwin principle, the inter-ligand dihedral angles between the dipyrrinato units were determined to be proportional to the size of the *α* ligands (R^1^) [186]. For bis(dipyrrinato)Ni(II) complexes **Ni-1** and **Ni-2** (Figure 23), either a distorted square planar geometry (R^1^ = H) or a distorted tetrahedral geometry (R^1^ = CH_3_, Ph) were observed [169,184,187,192,193]. A red-shift was observed in the UV-visible absorption spectra of **Ni-1b,c** and **Cu-1b,c** when the dihedral angle was increased by replacing the R^1^ = H substituents for R^1^ = CH_3_, yielding complexes **Ni-2b,c** and **Cu-2b,c** (Figure 24) [184].

Similar distorted geometries were observed for bis(dipyrrinato)Cu(II), -Co(II) and -Zn(II) complexes [15,79,108,169,185,189,190,194,195]. In general, Zn(II) complexes prefer a tetrahedral configuration bis(dipyrrinato)Zn(II) complexes typically show a dihedral angle of 90° [169]. Pd(II) complexes were shown to exhibit much smaller distortions from square planar (**Pd-1** and **Pd-3** compared to **Ni-1a,c** and **Cu-1a,c**). From these and other results it can be concluded that the order of deviation from the tetrahedral angle caused by *α*-substitution is: Zn(II) < Pd(II) < Ni(II) < Cu(II) [108]. The impact of the geometry on the photophysical properties of (dipyrrinato)Ni(II) and -Cu(II) complexes has not been studied yet, besides UV-Vis absorbance, to the best of the authors’ knowledge. It has been shown that the geometry has an influence on the magnetic properties (and thus the spin-state) of bis(dipyrrinato)Ni(II) complexes. Typically, Ni(II) square planar complexes are diamagnetic (singlet ground state) and tetrahedral complexes are paramagnetic (triplet ground state). Complexes in between those geometries can be either diamagnetic or paramagnetic (or exhibit a temperature-dependent equilibrium), depending on the dihedral angle and other substituents [15,169,184,187,191]. To illustrate, complex **Ni-1e** with a dihedral angle of 49.6° is paramagnetic, while complexes **Ni-1a,b** with a dihedral angles of 38.5° are diamagnetic [169,187]. These results shows that a small change in dihedral angle (11.1°) can already induce a change of spin in the ground state. Increasing the size of substituents on the *α*-positions (R^1^ = CH_3_, Ph), and thereby increasing the dihedral angle has been shown to result in paramagnetic states in some cases [169,184]. It is suggested that to tune the complexes towards displaying diamagnetic ground states by choosing a specific set of substituents at different positions of the dipyrrinato ligand. The influence of the meso-substituents was investigated by observing changes in the UV-Vis absorption spectrum upon changing the aromatic meso-substituent from tolyl to C_6_H_4_CN (Figure 24) [184]. For **Ni-1b**
**→ Ni-1c** and **Ni-2b**
**→ Ni-2c** a red shift was observed (14 nm and 2 nm, respectively). Similar red-shifts were observed for their Cu(II), Co(II) and Zn(II) analogues (**Cu-1bc, Cu-2bc**, **Co-2bc**, and **Zn-3cd**; see Table A4, Appendix A) [169,184]. Usually, Ni(II) and Cu(II) complexes are non-luminescent, which could be a problem for application as PS’s for PDT. On the contrary, (dipyrrinato)Cu(II) complexes **Cu-1a** and **Cu-3a** did display weak fluorescence [196]. Complex **Cu-1a** weakly emits around 500 nm and complex **Cu-3a** at 570–630 nm in several solvents. The fluorescence quantum yield of **Cu-3a** was estimated to be 0.001 in butyronitrile. Karges et al. investigated two iodo-substituted homoleptic bis(dipyrrinato)Ni(II) and -Cu(II) complexes (**Ni-3** and **Cu-4**, Figure 23) for potential PDT application [15]. These are direct analogues of **Zn-4g** (R^2^ = I), which was already shown to have potential for PDT [79]. Both complexes adopt distorted geometries in between square planar and tetrahedral (Figure 25). Compound **Ni-3** was found to be diamagnetic (dihedral angle of 81.0°), while **Cu-4** was determined to be paramagnetic (dihedral angle of 65.1–69.0°). Both complexes had strong absorption in the green area of the electromagnetic spectrum (around 525 nm), which is relatively red-shifted compared to other TM complexes (Figure 26). The absorption profiles were almost identical, with slightly stronger absorbance for both peaks of **Ni-3**. Photophysical studies demonstrated that both complexes were poorly emissive and scarcely generated singlet oxygen upon irradiation. No luminescence signal was detected for **Ni-3** and **Cu-4** displayed a low emission signal with a fluorescence quantum yield of 0.001. The biological evaluation revealed that they had low dark cytotoxicity in non-cancerous retinal pigment epithelium and human cervical carcinoma (HeLa) cells (IC_50_ = 157.1–185.3 µM); however, a negligible effect was observed on the cell viability upon irradiation (at 510 nm, 20 min, 5.0 J cm^−2^) The authors of this work suggest a follow-up investigation of α-unsubstituted analogues of **Ni-3** and **Cu-4**. It may be possible that removing strain at these positions, and the expected decrease in dihedral angles, allows for a new channel towards improving the photophysical properties of these complexes for PDT. The influence of the geometry of bis(dipyrrinato)Ni(II) and -Cu(II) complexes on their photophysical properties is yet unknown, and such a study may provide a starting point to such investigations.

Nishihara and co-workers synthesized heteroleptic (dipyrrinato)Ni(II) and -Cu(II) complexes **Ni-4** and **Cu-5** (Figure 23) [197]. X-ray crystallography showed that both complexes adopted tetrahedral geometries, due to the bulky α-substituents on one of the dipyrrinato ligands (Figure 27). Both complexes were non-emissive, while the heteroleptic and tetrahedral Zn(II) complexes mentioned before were emissive [165]. The structures were analyzed by X-ray crystallography, cyclic voltammetry, chronocoulometry, and UV-Vis absorption spectroscopy. No data with regards to PDT have been reported. However, this topic is relatively new, and we believe significant progress can still be made. For example, a similar study can be carried out on different substituent effects on their photophysical parameters, similar to studies of bis(dipyrrinato)Zn(II) complexes. It is suggested to use the earlier mentioned knowledge to design better suitable bis(dipyrrinato)Pd(II) complexes. Besides that, results for bis(dipyrrinato)Zn(II) complexes showed that introducing dissymmetry can drastically influence the photophysical properties. Therefore, a similar approach could be used to steer towards a higher triplet generation by choosing different substitutions on each of the two dipyrrinato ligands, similar to what was suggested before for bis(dipyrrinato)Zn(II) complexes.

#### 5.1.5. Further Comparisons between d-Block Metal Dipyrrinato Complexes

In this section, several articles are discussed that directly compare dipyrrinato complexes based on different metal ions with each other. In this way, the influence of the metal-ion on the photophysical properties can be directly observed. First of all, Telfer and co-workers compared the photophysical properties of several (dipyrrinato)Ru(II), -Rh(III), -Pd(II) and -Co(III) complexes with each other (**Ru-2a**, **Ru-2b**, **Pd-5**, **Pd-6**, **Rh-1a**, **Rh-1b**, and **Co-1**) [67,128]. Exciton coupling was observed in the absorption spectra of **Rh-1a** but not in those of **Pd-5** and **Ru-2a** (Figure 13) [67]. This could be explained by the fact that **Ru-2a** consists of only two dipyrrinato ligands (lower symmetry) and is therefore expected to show weaker (possibly non-visible) exciton coupling in the absorption spectrum. Additionally, a null net transition dipole moment to the lower energy π-π* band was suggested to cause the absence of a split π-π* band in the absorption spectrum of **Pd-5**. **Ru-2a**, **Ru-2b**, **Pd-5** and **Co-1** were found to be non-emissive and it was suggested that the low-lying, metal-centered (MC) d-d excited states could provide efficient non-radiative deactivation pathways and cause low emission yields. On the other side, Rh(III) analogues of these complexes (**Rh-1a** and **Rh-1b**) appeared to be luminescent (although with a small quantum yield). It was suggested that the ligand field splitting of Rh(III) these states can become inaccessible, thus removing some non-radiative deactivation pathways [67]. Guseva et al. investigated the influence of the metal on the properties of several dipyrrinato complexes (**Zn-11**, **Hg-1**, **Co-3**, **Ni-5** and **Cu-6**, Figure 28) compared to the free ligand [198]. Such a comparison gives an insight in how the metal can influence the photophysical properties. The d-metal ion exerts an auxochromic effect on the *π*-system of the ligand, as evidenced by a red-shift of the first absorption band, compared to the free ligand. In addition, the polarization of the dipyrrinato *π*-system followed an ascending order Zn(II) < Hg(II) < Co(II) < Ni(II) < Cu(II). Strikingly, this order almost resembles the Irving-Williams series characterizing the stability of metal complexes (Co(II) < Ni(II) < Cu(II) > Zn(II)), especially when Zn(II) become less stable than Co(II) complexes due to the possibility for *π*-back-bonding with the coordinated N double-bonded with a carbon [199]. These results, as well as other articles, show that dipyrrinato ligands can form complexes with both Co(II) and Co(III) ions [195]. In an earlier paper it was shown that similar (dipyrrinato)Co(II) complexes were unstable, difficult to isolate, air-sensitive or they were transformed into tris(dipyrrinato)Co(III) complexes similar to **Co-1 [198,200]**. In a similar way, Das and Gupta have reported the synthesis, photophysical and electrochemical properties of triphenylamine substituted dipyrrinato metal complexes. Amongst these complexes where five homoleptic dipyrrinato complexes of Ni(II), Co(II), Pd(II), Zn(II) and In(II) (Figure 28) [201]. The photophysical properties of **Co-4**, **Ni-6**, **Pd-9**, **Zn-12** and **In-1** (Figure 28) were determined in five different solvents. Similar to bis(dipyrrinato)Zn(II) complexes, the rotational freedom of meso-substituents has an influence on the photophysical properties. A sterically more hindered meso-aryl group was shown to enhance emission [162]. Das and Gupta further investigated the influence of the bulkiness of the meso-aryl group by comparing the photophysical properties of bulky triphenylamine substituted dipyrrinato complexes with their phenyl substituted analogues. **In-1** contains a trivalent metal ion and complexes with three dipyrrinato ligands, whereas the other complexes are divalent and complexes with two dipyrrinato ligands. Similar tris(dipyrrinato)In(III) complexes will be discussed later. All five complexes showed an intense absorption band in the region of 430–480 nm in toluene, corresponding to the *π−π** transitions from the dipyrrinato ligand. **Pd-9** showed only one band in DCM at 435 nm, whereas **Pd-5** showed two bands at ca. 380 nm and 460 nm. The absorption spectra of **Ni-6**, **Pd-9**, **Zn-12** and **In-1** were blue-shifted compared to their phenyl substituted analogues and showed a red-shift upon increasing the solvent polarity. Complexes **Pd-9**, **Zn-12** and **In-1** exhibited red shifted emission maxima and reduced quantum yields compared to their phenyl substituted analogues. **Co-4** and **Ni-6** were non-fluorescent. Fluorescence quantum yields, Stokes shifts and singlet state lifetimes were measured for the other complexes. The fluorescence quantum yields of **Pd-9**, **Zn-12** and **In-1** ranged between 0.013 and 0.025 in ethyl acetate. It was suggested that the yield was relatively low due to non-radiative decay processes involved; however, these were not investigated further. Stokes shifts of **Pd-9**, **Zn-12** and **In-1** were found to be in the range of 3648–7000 cm^−1^. Emission data showed that the rates of non-radiative decay processes were higher than the radiative processes for all complexes.

### 5.2. p-Block Dipyrrinato Complexes

The p-block of the periodic table (Group 13–18) contains metals, metalloids, and non-metals. Especially complexes with elements from Group 13 have been widely investigated in photomedicine. The best known dipyrrinato complexes with elements from this group are BODIPY dyes, with a boron center ion, due to their relatively facile preparation and functionalization [12,13,14,46]. Likewise, the group 13 elements aluminum (Al), gallium (Ga) and indium (In) have been shown to form complexes with dipyrrinato ligands. Their photophysical properties and application in PDT will be discussed further in the following sections [3,83,113,125,202,203,204,205]. Dipyrrinato complexes with other p-block elements have been less described in the literature. Heteroleptic dipyrrinato complexes have been synthesized with Tl(III), Sn(II) and Sn(IV) as the central ion [206,207,208]. The synthesis of a silicon mono(dipyrrinato) complex was attempted, which surprisingly yielded other structures with 2-fold deprotonated 1,1-bis(pyrrol-2-yl)ethene as a ligand [209]. From group 15, only phosphorous has been shown to be able to coordinate with both regular dipyrrinato ligands and aza-dipyrrinato ligands [113,121,210,211]. Additionally, N_2_O_2_-type ligands were shown to form complexes with B, In, Ga, Si, Ge, Sn and Al [114,116,117,118].

#### 5.2.1. Dipyrrinato-Ga(III) and -In(III) Complexes

Dipyrrinato ligands have been shown to be able to form complexes with both gallium (Ga) and indium (In) metals. Several tris(dipyrrinato)Ga(III) and -In(III) complexes (Figure 29) and their absorption spectra (Figure 30) have been reported [3,83,125,203]. Additionally, a few mono(dipyrrinato)Ga(III) complexes have been investigated [113,202]. The first preparation of luminescent dipyrrinato complexes with these metals (**Ga-2**, **Ga-3** and **In-3**) was reported in 2006 by Cohen and co-workers, who also reported characterization by X-ray crystallography, absorption and fluorescence spectroscopy, TCPC, and computational methods [3]. The products were unstable during column chromatography, which led the researchers to find a synthetic procedure which could circumvent this chromatographic purification of the final product. Hence, they first prepared and purified the bis(dipyrrinato)Cu(II) analogue, which gave less problems. The copper was removed from the pure Cu(II) analogue by adding KCN to the solution, after which pure dipyrrinato ligands could be separated and isolated in high yield using a simple extraction. In a subsequent step, the complexes **Ga-2** and **In-3** were prepared by mixing 3 equivalents of ligand with 1 equivalent of the appropriate metal salt in a methanol/acetonitrile mixture. It was suggested that this strategy could also be applicable to other dipyrrinato complexes that may be difficult to purify. X-ray crystallography determined that the structures of tris(dipyrrinato)Ga(III) complex **Ga-2** was very similar to that of tris(dipyrrinato)In(III) complex **In-3**.

The absorption spectra of **Ga-2**, **Ga-3** and **In-3** were very similar to each other (Figure 31b) and display an intense split band with maxima at ca. 450 and 500 nm. At shorter wavelengths a high-energy transition and vibronic structure was observed. This feature was not visible in spectrum of the **Zn-2g** analogue (see Figure 16) [164]. Both **Ga-2** and **In-3** were found to be fluorescent (λ*_f_* = 528 and 522 nm, respectively), although less fluorescent than their BODIPY or (dipyrrinato)Zn(II) analogues. The fluorescence quantum yield was determined to be higher for **In-3** than **Ga-2** ($\varphi_{f\;}$= 0.074 vs. 0.024 in hexanes). However, compared to **Zn-2g** ($\varphi_{f}$ = 0.36 in toluene) both complexes are considerably less luminescent. This indicates that non-radiative decay pathways (such as IC and ISC) are dominant. The Stokes shifts (∆ṽ = 1113–1220 cm^−1^) were found to be much larger than for **Zn-2g** (∆ṽ *≈* 574 cm^−1^). It was concluded that the most prominent electronic transitions in these complexes are ligand-centered. **Ga-3** was not notably luminescent and therefore was not investigated further. Using TCPC the fluorescence lifetimes of **Ga-2** and **In-3** were obtained in several solvents including toluene, DCM and mixed hexanes (see Table A5, Appendix A). In all solvents tested, except for methylcyclohexane and hexanes, the complexes underwent degradation, possibly via solvolysis. Furthermore, **Ga-2** and **In-3** were more stable in noncoordinating solvents than **Zn-2g**. The fluorescence life-time $(\tau_{f})\;$of **Ga-2** was found to be larger than that of **In-3** (3.75 ns vs. 1.93 ns) and slightly larger than that of **Zn-2g** (3.51 ns) in a solvent mixture of hexanes. Although triplet states were not directly investigated, the detection of ’transients’ via flash photolysis was mentioned. Ground-state bleaching and excited-state transient absorption were detected in both **Ga-2** and **In-3**, similar to **Zn-2g**. The transients detected had lifetimes of 300–320 ns in air, 56–62 ns in 1 atm. O_2_ and 45.000–80.000 ns in the argon-purged case. This indicates sensitivity to oxygen, which was also observed for **Zn-2g**, and might indicate possible applications as PSs. According to the authors, the detection of the O_2_-sensitive ’transients’ could imply a partial decay of the excited state compound towards a triplet state. The ’transient’ features were more intense for **In-3**, which is expected considering that In(III) is heavier than Ga(III) and therefore a larger internal HAE is observed. Factors that play a key role and affect the absorption and emission properties include the metal center but also the number and nature of the coordinated ligands. The homoleptic tris(dipyrrinato)In(III) complexes **In-3** and **In-6** were emitting less ($\varphi_{f,3}$ = 0.074 in hexane and $\varphi_{f,6}$ = 0.028 in toluene) than their heteroleptic analogues **In-4** and **In-5** ($\varphi_{f,4}$ = 0.41 and $\varphi_{f,5}$ = 0.34 in toluene) [3,203]. The UV-Vis absorption spectra of complexes **In-4**, **In-5** and the **In-6** (Figure 30) show differences that can be attributed to different exciton coupling between the ligands, similar to what was seen for **Rh-1a** (Figure 13) [67]. Peaks found in the range of 450–510 nm can be attributed to the R = H dipyrrinato ligand (D1), evidenced by the absence of this peak for **In-6**. Bands in the range of 500–600 nm can be attributed to emission from the R = TA dipyrrinato ligand (D2). This is supported by a higher intensity for **In-5** with two D2 ligands compared with **In-4** with only one. Heteroleptic **In-4** (green) displayed exciton coupling between the two D1 ligands, resulting in a sub-band at 500 nm. Similarly, heteroleptic **In-5** (yellow) displayed an absorption band at 546 nm with a shoulder that can be attributed to exciton coupling between the two D2 ligands. A more significant band splitting of the ^1^π-π* transition was observed for homoleptic **In-6** (red) due to exciton coupling between three identical D2 ligands. The emission spectra of **In-(4–6)** showed one band, independent of the excitation wavelength. In this case, the emission band resulted from emission of the ligand containing the TA groups. This behavior can be explained by energy transfer from ligand D1 to ligand D2, which results in less efficient CT. This suppression was more prominent in DCM than in toluene. Surprisingly, **In-4** displayed a larger fluorescence quantum yield than its TA-substituted BODIPY analogue **B-4** ($\varphi_{f}\;$= 0.41 vs. 0.35 in toluene). This is an important observation, as it indicates that (heteroleptic) (dipyrrinato)metal complexes could emit more brightly than BODIPYs, contrarily to what was previously thought. To conclude, CT for heteroleptic tris(dipyrrinato)In(III) complexes can be suppressed and higher fluorescence yields can be obtained.

Several applications of Ga(III) and In(III) complexes have been explored, e.g., as possible building blocks in self-assembling antenna pigments, light-harvesting complexes, or luminescent MOFs [125]. More recently, applications as phototherapeutic agents have been studied. In one study, homoleptic tris(dipyrrinato)Ga(III), -In(III) and -Fe(III) complexes were investigated (Figure 29) [83]. The absorption spectra of the Ga(III) and In(III) tris(dipyrrinato) complexes showed two distinct absorption bands at ~450 and ~515 nm whereas the spectrum of the Fe(III) complex these bands are evidently broader. The cytotoxicity of selected complexes **Fe-2,3** and **Ga-5,6** in the absence and presence of light was evaluated in human epidermoid carcinoma (A253), human epithelial carcinoma (A431), human oral adenosquamous carcinoma (CAL27), and colorectal adenocarcinoma (HT29) cell lines, whilst the mouse fibroblast cell line L929 was tested as a non-tumorous cell line. Lastly, the bacteria photoinactivation was evaluated with **Fe-2,3** and **Ga-5,6** against the Gram-positive bacterium *S. aureus*. Overall, the phototoxic effect against tumor cells and against *S. aureus* was predominantly observed with the glycosylated gallium(III) complexes (**Ga-5** and **Ga-6**) and the activity was dependent on the metal and the presence of carbohydrate unit. The described behavior of the bis(dipyrrinato)Ga(III) and -In(III) complexes shows similarity with each other and with that of bis(dipyrrinato)Zn(II) complexes. These observations are not too surprising, as Ga(III), In(III) and Zn(III) are all *d*^10^ metals with completely filled d-orbitals and are expected to have similar properties. A theoretical study of complexes **Ga-1** and **In-2** showed indeed very similar energy level diagrams and frontier orbitals as their bis(dipyrrinato)Zn(II) analogue **Zn-1** (Figure 31a) [3]. It was shown that for all three complexes the most likely low-energy electronic transitions are ligand centered π−π* transitions. Additionally, the UV-Vis spectra of **Ga-2** and **In-3** looked almost identical (Figure 31b) [3].

This similarity allows for the use previous knowledge on tuning bis(dipyrrinato)Zn(II) complexes as inspiration to design new (dipyrrinato)Ga(III) and -In(III) complexes. One especially noteworthy observation is the similarity of the behavior of heteroleptic tris(dipyrrinato)In(III) complexes **In-4** and **In-5** to that of heteroleptic Zn(II) analogues **Zn-6** and **Zn-7**, all of which showed improved fluorescence quantum yields compared to their homoleptic equivalents. Similar investigations for heteroleptic tris(dipyrrinato)Ga(III) complexes are suggested, which also involve *d*^10^ metals. Additionally, the introduction of dissymmetry as a tool for enhancing triplet generation (instead of fluorescence suppression) in these complexes, similar to what was seen for **Zn-8**, could be investigated further. Less is known about other *d*^10^ bis(dipyrrinato) complexes such as tris(dipyrrinato)Cd(II) and -Hg(II) complexes; however, a few examples have been mentioned in the literature [182,185,198,212,213,214,215,216]. For instance, Cd(II) ions in bis(dipyrrinato)Cd(II) complexes are able to bind with one or two additional ligands, which Zn(II) cannot [214]. Bis(dipyrrinato)Hg(II) complexes were shown to have potential as agents for the mercuration of benzene and chloroform [198]. Besides the tris(dipyrrinato) complexes, a few mono(dipyrrinato)Ga(III) complexes (GaDIPYs) have been described (Figure 32) [113,202]. Due to the similarity of gallium to boron, GaDIPYs are expected to display similar bonding geometries and photophysical properties as their BODIPY analogues. Substitution of the B-atom by a heavier main group element such as Ga is expected to influence its optical and electronic properties greatly [217,218,219,220]. An extra benefit of GaDIPYs over BODIPYs could be the possibility for biomedical radiolabeling, similar to other Ga-based compounds [113]. Recently, Wan et al. reported the synthesis of the stable monomeric gallium chelate **Ga-8**, by using an intermediate lithium-dipyrrin salt [113]. This new preparation technique showed the possibility of gaining access to heavier main group analogues of BODIPY. **Ga-8** resembled the highly fluorescence profile and the photophysical properties of the BODIPY analogues along with a similar tetrahedral geometry. A slight blue shift was observed for both the absorption and emission for exchanging the boron for the gallium at the center. **Ga-8** was shown to have highly luminescent properties ($\varphi_{f}$ = 0.82 in DCM, 0.91 in toluene) [113]. Usually, dipyrrin complexes are considered as poor emissive chelates; however, this example shows this is not a strict rule [5]. In contrast, tris(dipyrrinato)Ga(III) complex **Ga-1** showed poor fluorescence emission ($\varphi_{f}$ = 0.024 in hexane). In a separate study DFT was used to determine the influence of replacing the fluoride ligands with C_20_-fullerene on **Ga-9** on the photophysical properties was investigated (Figure 32) [202]. It was found that the presence of C_20_ in **Ga-10** caused a small decrease in the LUMO energy and a large increase in the HOMO energy compared to **Ga-9**. This change in frontier orbital energies resulted in a significant decrease in the HOMO-LUMO bandgap (3.18 eV to 1.27 eV). This was due to the formation of a new high-energy level (mostly on C_20_) acting as the HOMO. Finally, it was determined that charge transfer occurred from the DIPY to C_20_. Similar observations were made for their boron and aluminum analogues.

#### 5.2.2. Dipyrrinato-Al(III) Complexes

Aluminum (Al) belongs to Group 13 elements of the periodic table together with boron, gallium, indium, and thallium. Aluminum is the third most abundant element found in the earth’s crust (8.3% by weight) and is exceeded only by oxygen (45.5%) and silicon (25.7%) [221]. Besides its abundance, aluminum is cheap compared to other metals mentioned in this work. Aluminum finds a wide range of uses in material, pharmaceutical sciences, and food industry [222,223]. An example of aluminum drugs in PDT is a sulfonated aluminum phthalocyanine (*Photosens^®^*), which has been clinically approved in Russia for the treatment of lung, liver, breast, skin, and gastrointestinal cancer [224]. With reference to aluminum dipyrrinato complexes only little advances and few reports have been made to date. Such complexes could however be of interest as alternative PSs due to the resemblance of aluminum to boron. It is envisaged that the triplet generation of (dipyrrinato)aluminum complexes can be enhanced with similar structural modifications as those established for BODIPYs [13,14].

Giannopoulos et al. described the synthesis of monomeric (dipyrrinato)Al(III) (AlDIPY) complexes bearing mesityl substituents at the *α*-pyrrolic positions **Al-(1–11)** (Figure 33) [204,205]. The introduction of bulky aryl groups at the *α*-position of the pyrrole ring results in a steric hindrance which stabilize the reactive Al(III) center towards the formation of the mono dipyrrin aluminum complexes. Complexes **Al-(1–4)** were prepared via reactions with the appropriate organo-aluminum reagents. **Al-(4–6)** could also be prepared via a reaction with a (dipyrrinato)Lithium salt via salt metathesis, a more practical approach. Compound **Al-6** was used as a precursor for **Al-8** and **Al-9**, also via salt metathesis. **Al-7** was prepared by reacting the non-complexed dipyrrinato ligand with [Al(N(CH_3_)_2_)_3_)]_2_. The preparation of **Al-9** turned out to be tedious, as this reaction yielded a mixture with **Al-7** which was difficult to separate. All complexes were studied by single crystal X-ray crystallography which showed the dipyrrin-ligand to be non-planar for bulky R^1^ and R^2^ ligands (such as I, O*^t^*Bu and N(CH_3_)_2_). With smaller R^1^ and R^2^ ligands (such as Cl) the dipyrrinato ligand was observed in a planar configuration. Whilst the photophysical properties were beyond the aim of the work, it was noted that compound **Al-1** was fluorescent (green) whilst complexes **Al-(2–4)** were non-emissive. Complexes **Al-(2–4)** were stable to air or moisture while **Al-1**, **Al-5** and **Al-6** were moisture sensitive, with dihydride **Al-1** resulting in the formation of oxo-bridged aluminoxane **Al-11** upon reaction with water. **Al-7** and **Al-8** were applied as catalysts for the ring-opening polymerization of ɛ-caprolactone in toluene. These catalysts were also shown to be water-sensitive, which could induce catalyst deactivation. Ikeda et al. reported N_2_O_2_-type (dipyrrinato)Al(III) complexes (**Al-12** and **Al-13**) which were found to be more stable than regular AlDIPYs [119]. These complexes displayed an absorption maximum at ca. 600 nm and exhibited moderate to high fluorescence quantum yields (see Table A6, Appendix A). Specifically, the mesityl derivative showed higher fluorescence quantum yield than the phenyl derivative due to the restricted rotation of the mesityl group ($\varphi_{f,12}$ = 0.23 and $\varphi_{f,13}$ = 0.72 in Tol:MeOH, 99:1). The fluorescence quantum yield of **Al-13** was identical to that of its BODIPY-analogue (**B-5**). The aluminum coordination in N_2_O_2_-type AlDIPY complexes adopted a square planar geometry, which allowed for extra octahedral coordination. Additionally, the phenolate oxygen atoms bonded to the aluminum center can facilitate further interactions. This was illustrated by the formation of heterometallic Al(III)-dipyrrinato-Zn(II) adducts (**Al-(14–16)**) [119]. The fluorescence intensity and wavelength of **Al-12** and **Al-13** significantly changed upon the addition of zinc salts ($\varphi_{f,14}$ = 0.55, $\varphi_{f,15}$ = 0.56, $\varphi_{f,16}$ = 0.83 in Tol:MeOH, 99:1) [119]. This was probably caused by the enhanced rigidity of the dipyrrin framework after chelation, reducing the energy loss via radiationless decay. The crystal structure of **Al-14** is shown in Figure 34. The boron–complex did not form this additional Zn(II)-salt coordination, most likely due to a distorted tetrahedral geometry instead of a square planar geometry [225]. Another explanation could be a lower negative charge on the oxygen atoms. Indeed, it was found that the Al-O bonds were more negatively charged than those of the B-O bond, which may prevent the coordination to Zn(II) ions [226]. Another report of aluminum dipyrrin complexes investigated the binding ability of AlDIPY complexes with alkaline earth ions (Ca^+2^, Mg^+2^, Ba^+2^, Sr^+2^). Both monomer [**Al-17**·(CH_3_OH)(H_2_O)] and dimer [**Al-18**·(CH_3_OH)_4_] aluminum complexes worked as selective receptors for alkaline earth metal ions via selective binding in aqueous media [227]. It has been demonstrated that lacking of a pyrrolic substitution at *α*-positions allows for the formation of 1:3 metal to ligand complexes of trivalent metals [3,83,125,129,131,203]. The common oxidation state of aluminum is +3; hence, in the absence of *α*-substitution the complex can adopt an octahedral coordination geometry. A Japanese patent by Toguchi et al., 20 years ago, reported the structures of tris(dipyrrinato)Al(III) complexes **Al-(19–22)** (Figure 33). Apart from this patent, which was directed for an application towards optoelectronics, there is no chemical neither photophysical characterization of such complexes [228]. As a courtesy to the reader we provide the d orbital occupancy [181] of the metal ions discussed in this work in Table A7.

## 6. Conclusions and Future Prospects

Tetrapyrrolic complexes are used as photosensitizers; however, several disadvantages still remain and therefore the search for alternative photosensitizers is a growing research area. Notably transition metal complexes have shown the ability to overcome some of these disadvantages. The metal introduces an internal heavy atom effect, which can enhance the triplet generation rate. Additionally, the metal allows for flexibility in ligand coordination, facilitating precise tuning of the photophysical properties. Using TM complexes as PSs for PDT is still relatively new. Inspired by the promising photophysical properties of BODIPYs, the dipyrrinato system is now attracting interest as a ligand for the development of photosensitizers for PDT. By discussing the existing literature on this topic, we explored the metal coordination effect on the photophysical properties of dipyrrinato complexes, with an emphasis on properties relevant for their application as photosensitizers for PDT. This work addressed the potential for further progress in this direction and several arenas for future research were suggested. All things considered, the dipyrrin ligand has been shown to be a promising candidate in the search for alternative transition metal photosensitizers.

This review sheds light on advances in our understanding of the photochemical and photophysical properties of (dipyrrinato)metal complexes and their potential in PDT. It is evident that further research is required to elucidate the detailed effects of metal coordination and substituents on the photophysical properties. First, taking into consideration previous research on complexes of 2nd and 3d row Group 7–9 metals with other types of ligands (mostly polypyridyl), it would be interesting to see whether the research findings on these complexes could be replicated for dipyrrinato complexes. For example, polypyridyl complexes based on Ru(II), Rh(III), Ir(III), Os(II), and Re(I) are attracting attention as PDT photosensitizers [25,37,95,96,106]. Of these metals, only a few (dipyrrinato)Ru(II), Rh(II), -Ir(III) and Re(I) complexes have been investigated as potential PSs in PDT [17,18,130]. Additionally, a few heteroleptic (dipyrrinato)Rh(III) complexes have shown anticancer activity via interactions with DNA; however, no applications for PDT have been described yet [149,153,154]. Osmium (Os)-based dipyrrinato complexes have not yet been developed. Further research could focus on assessing the similarity of the photophysical behavior of these dipyrrinato complexes to complexes with similar ligands and eventually on the application of these types of complexes to PDT. 2^nd^ and 3d row Group 7–9 metal ions discussed in the former paragraph are relatively expensive, not very abundant and less biologically relevant than other possible metals in the periodic system. First row transition metal ions such as Fe(II), Fe(III), Co(II), Co(III), Ni(II), Cu(II), and Zn(II) may provide less expensive, more abundant and more biologically compatible alternatives to currently used PSs in the field of PDT. A few articles have already been dedicated to the application of (dipyrrinato)Ni(II), -Cu(II) and -Zn(II) complexes as PSs for PDT [15,79]. Their potential is further emphasized by studies on the PDT application of complexes with the related polypyridine ligands. For example, Fe(II) complexes with three π-extended bipyridine ligands and Fe(II) complexes with three 4,7-diphenyl-1,10-phenanthroline ligands have been investigated as PSs for PDT [229,230]. Lastly, related Cu(II) polypyridine, such as terpyridine, have received increasing attention as anticancer drug candidates [231]. Bis(dipyrrinato)Zn(II) complexes were shown to display unique photophysical properties when complexed with dipyrrin ligands, involving the formation of non-emissive LLCT states and efficient generation of triplet excited states. The substituent effects on the photophysics of these complexes have been thoroughly studied, including but not limited to substituents on the *α-*, β*-* and meso-positions and introducing dissymmetry. A yet unexplored area is the use of dissymmetry in heteroleptic bis(dipyrrinato)Zn(II) complexes, specifically to enhance triplet generation. Until now, this approach has been only applied with the goal of increasing the fluorescence quantum yields compared to their homoleptic analogues [120,165,177]. One example has shown that attaching electron donating substituents on the β-positions of one of the dipyrrinato ligands can decrease the fluorescence yield; however, the benefit for improving ISC and enhance the triplet generation has not been mentioned [165]. Considering multiple suggestions for new research areas that have been proposed in this work, it is anticipated that significant progress can still be made for the application of bis(dipyrrinato)Pd(II), -Cu(II) and -Ni(II) complexes as photosensitizers for PDT. Except for one article, this direction has not been thoroughly investigated yet to date [15]. It is expected that great steps can be made for these three types of dipyrrinato complexes by using similar ISC enhancement techniques as applied for bis(dipyrrinato)Zn(II) complexes (e.g., structural modifications, introducing dissymmetry). It is unknown yet whether such modifications lead to comparable results on the photophysical properties, and therefore this direction is worthwhile to pursue. The effects of Cu(II) and Ni(II) on porphyrin type compounds should be considered as well. An extension of such a project could be made towards (dipyrrinato)Pt(II) complexes. Preliminary results show that iodo-free homoleptic (dipyrrinato)Pd(II) complexes have higher singlet oxygen yields than their iodo-substituted Zn(II) analogues [178]. For bis(dipyrrinato)Cu(II) and -Ni(II) complexes in specific, more knowledge on the substituent effects on the geometry, ground state magnetic properties and photophysical properties would be valuable. Especially the effect of the size of α-substituents on the dihedral angles, co-planarity and curvature of the dipyrrinato ligands is expected to be a significant contributor to the photophysical properties of these complexes. A starting point could be to prepare and characterize α-unsubstituted analogues of **Ni-3** and **Cu-4** and compare their photophysical properties [15]. An additional suggestion for research focus is enhancing ISC and triplet generation for p-block dipyrrinato complexes. No research has yet been dedicated to this topic, other than two brief mentions of tris(dipyrrinato)Ga(III) and -In(III) complexes [3,83]. Except for BODIPYs, it is therefore still unclear whether these complexes have any potential as photosensitizers for PDT. An example of a new research direction is investigating the possibility of enhancement triplet generation in AlDIPYs and other p-block mono(dipyrrinato) complexes by using similar techniques as developed for BODIPY’s such as structural modifications [232]. Due to their similarity, tuning the photophysical properties tris(dipyrrinato) complexes with Ga(III) and In(III) metal ions is expected to be possible in an identical way as has been performed for bis(dipyrrinato)Zn(II) complexes. One article already described the preparation of heteroleptic In(III) complexes, but the possible benefit of introducing dissymmetry as a tool towards enhancing triplet and singlet oxygen generation has not yet been discussed [203]. Aza-dipyrrinato complexes have shown beneficial properties relevant for PDT such as near infrared absorption and large Stokes shifts. Only a few articles described transition metal complexes based on aza-dipyrrinato complexes, indicating that metal dipyrrinato complexes have some advantages over their regular dipyrrinato analogues [48,49,120,121,210]. The potential use of aza-dipyrrinato complexes in PDT again requires further research; for example, using different metals, substituents and co-ligands. The field may benefit from using a theoretical approach. For example, by developing theoretical models based on previously obtained knowledge on the structures and photophysical processes of a large range of complexes. A prediction could be made on which structures (e.g., metals, ligands, substituents) structures may have potential for PDT as has been already shown for BODIPYs [232]. This avoids the synthesis, characterization and comparison of a large range of complexes, of which a large portion may never be applied as actual PSs for PDT. Taken all these aspects into account, it is clear that the dipyrrin ligand is a valuable candidate that can be combined with many transition metals in the search for new photosensitizers.

## Figures and Tables

**Figure 1 molecules-27-06967-f001:**
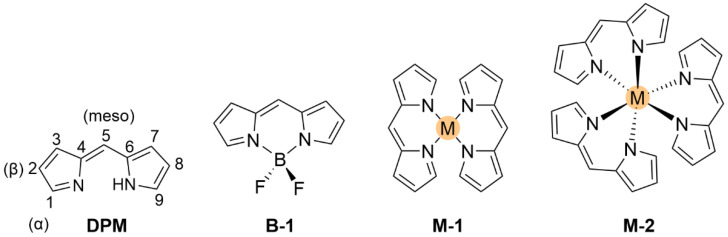
Chemical structures of a dipyrromethene (**DPM**), a mono(dipyrrinato) complex (BODIPY **B-1**), example of a bis(dipyrrinato)metal(II) (**M-1**) and a tris(dipyrrinato)metal(III) (**M-2**) complex. The atomic numbering scheme is given for **DPM**.

**Figure 2 molecules-27-06967-f002:**
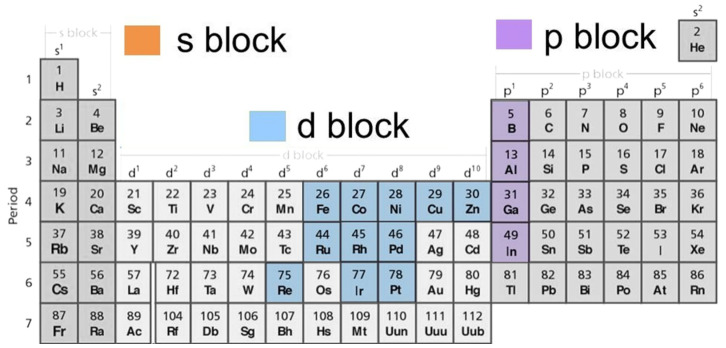
Periodic table visualizing the elements of which the dipyrrinato complexes are discussed in this work for their potential as PS’s for PDT (colored). Octahedral homoleptic tris(dipyrrinato) complexes discussed here, are formed mainly by Al(III) *d*^0^, Ga(III) *d*^10^, In(III) *d*^10^, Fe(III) *d*^5^, Co(III) *d*^6^ and Rh(III) *d*^6^. Tetrahedral or square planar bis(dipyrrinato) complexes are formed mainly by Co(II) *d*^7^, Ni(II) *d*^8^, Cu(II) *d*^9^, Zn(II) *d*^10^ and Pd(II) *d*^8^. Heteroleptic mono(dipyrrinato) complexes are formed mainly (but not exclusively) with B, Al(III) *d*^0^, Ga(III) *d*^10^, Ru(II) *d*^6^, Rh(III) *d*^6^, Pd(II) *d*^8^, Re(I) *d*^6^, Ir(III) *d*^6^ and Pt(II) *d*^8^. Note that the d orbital occupancy is given for every ion in this caption (see also Appendix A).

**Figure 3 molecules-27-06967-f003:**
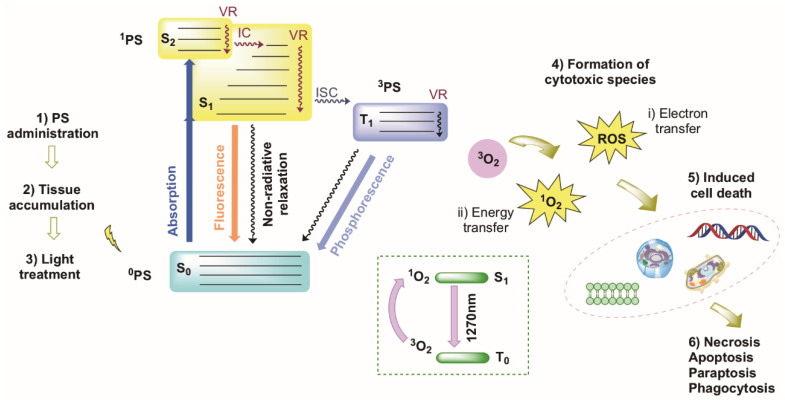
Simplified and generalized schematic Jablonski diagram showing the mechanism of PDT. Note, that the character and the energy of the long-lived state (here T_1_) can be modified by the metal ion complexation.

**Figure 4 molecules-27-06967-f004:**
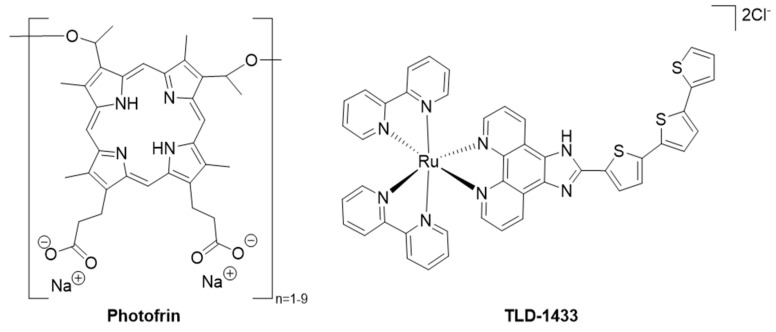
Structure of *Photofrin* (porfimer sodium) and the Ru(II)-complex *TLD-1433*.

**Figure 5 molecules-27-06967-f005:**
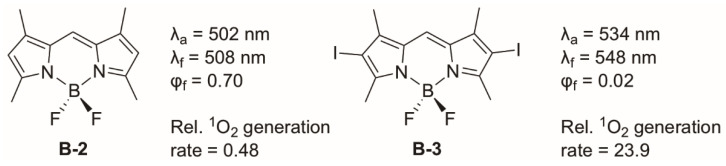
Change in photophysical parameters (in MeOH) for iodo-substitution at the β-positions of BODIPY **B-2** [55]. ^1^O_2_ generation rate was determined relative to methylene blue.

**Figure 6 molecules-27-06967-f006:**
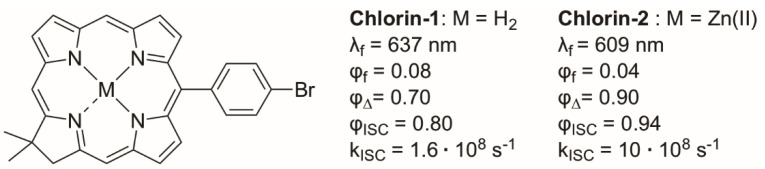
Zn(II) coordinated *gem*-dimethyl chlorin (**Chlorin-2**) and its free base counterpart (**Chlorin-1**). Experimental results shown in ethanol [63].

**Figure 7 molecules-27-06967-f007:**
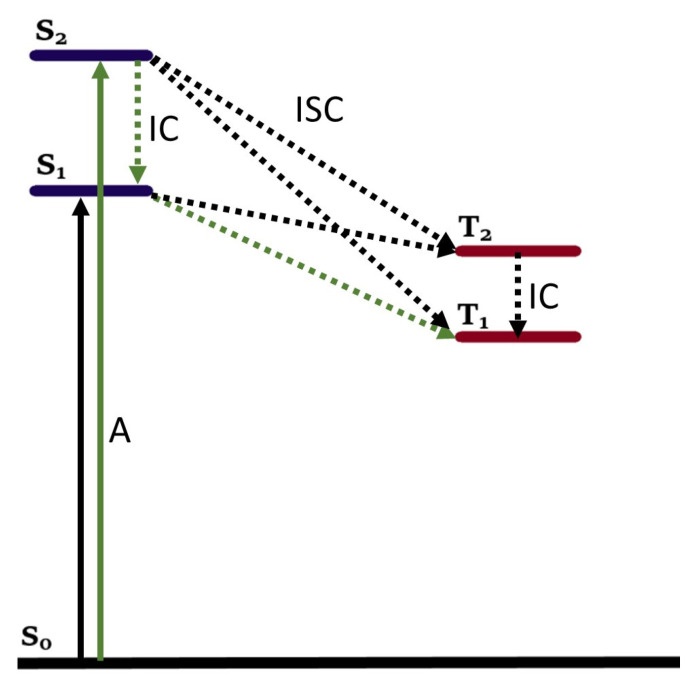
Energy levels involved in triplet generation. The most probable pathway is highlighted in green.

**Figure 8 molecules-27-06967-f008:**
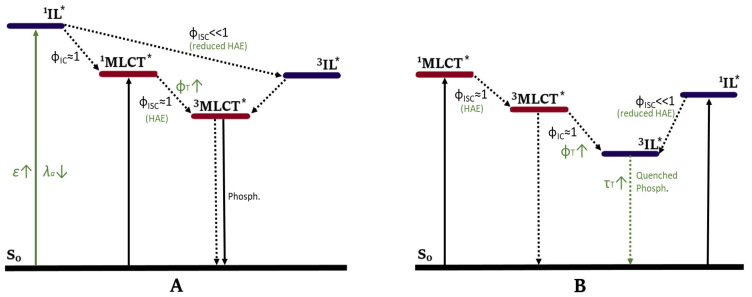
Energy level diagrams describing the photophysical processes of TM PSs with a visible light-harvesting ligand [25]. (**A**) Energy diagram for attaining strong absorption in the visible range. E(^3^IL) > E(^3^MLCT) [93] (**B**) Energy diagram for attaining a long-lived ^3^IL state. E(^3^IL) < E(^3^MLCT) [94]. For both diagrams the transition ^1^MLCT → ^3^MLCT is more efficient due to the HAE and the transition ^1^IL → ^3^IL is less efficient due to a reduced HAE. The excited state is denoted by a *.

**Figure 9 molecules-27-06967-f009:**
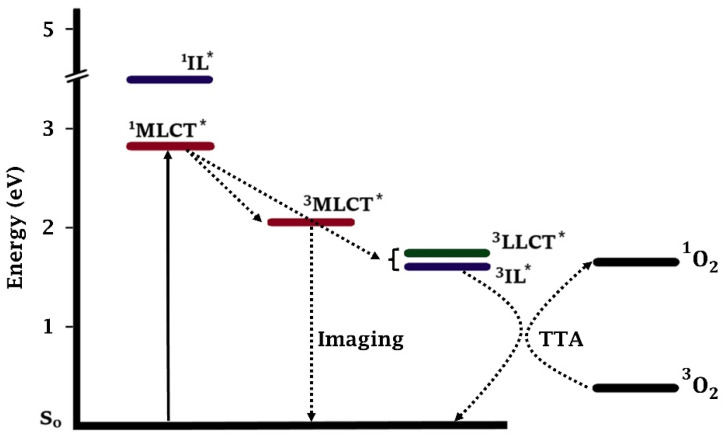
Energy level diagram describing the photosensitizing process of *TLD-1433* [37].

**Figure 10 molecules-27-06967-f010:**
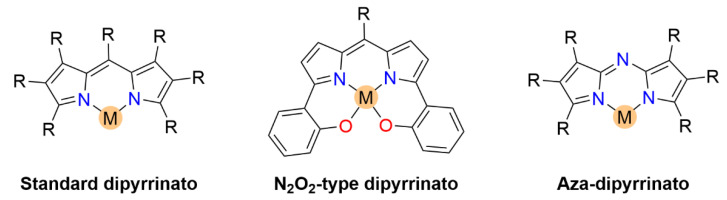
Three different types of dipyrrinato-based complexes with unspecified side groups (R). Nitrogen atoms are depicted in blue; oxygen atoms are depicted in red.

**Figure 11 molecules-27-06967-f011:**
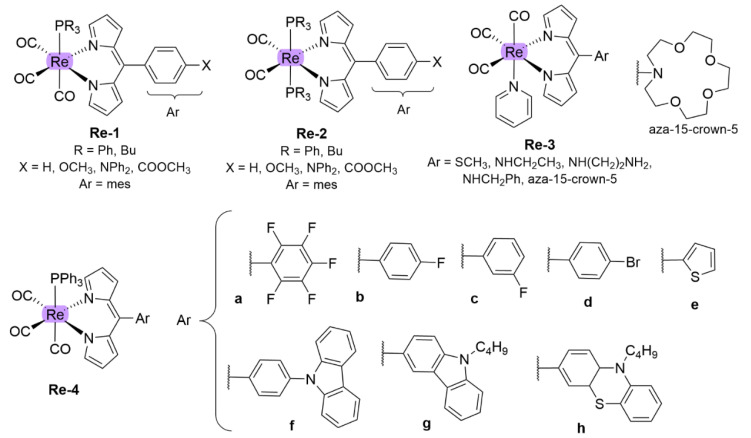
Examples of (dipyrrinato)Re(I) complexes reported in the literature.

**Figure 12 molecules-27-06967-f012:**
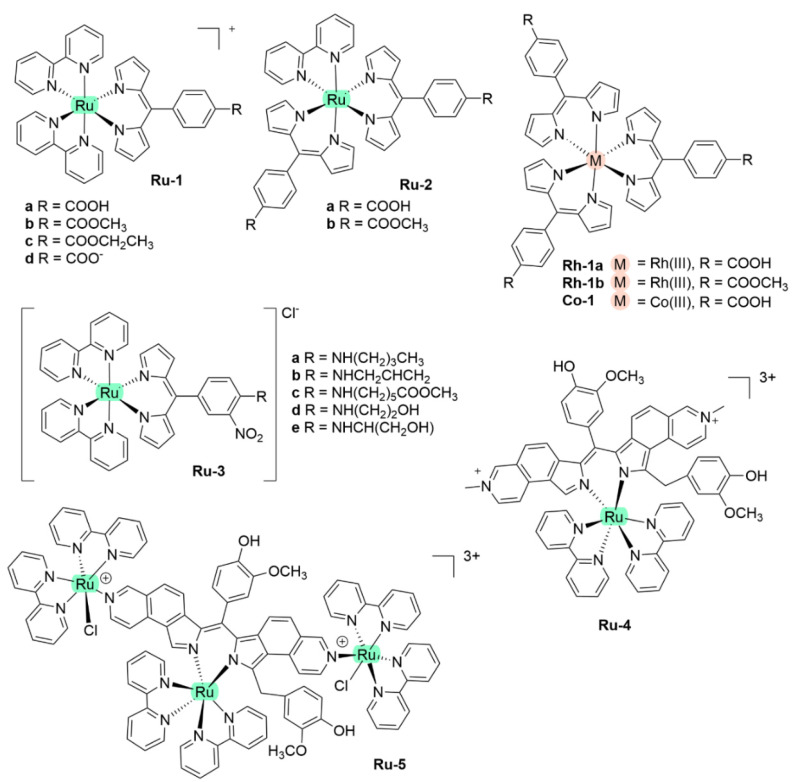
Examples of reported (dipyrrinato)Ru(II) and -Rh(III)-based complexes.

**Figure 13 molecules-27-06967-f013:**
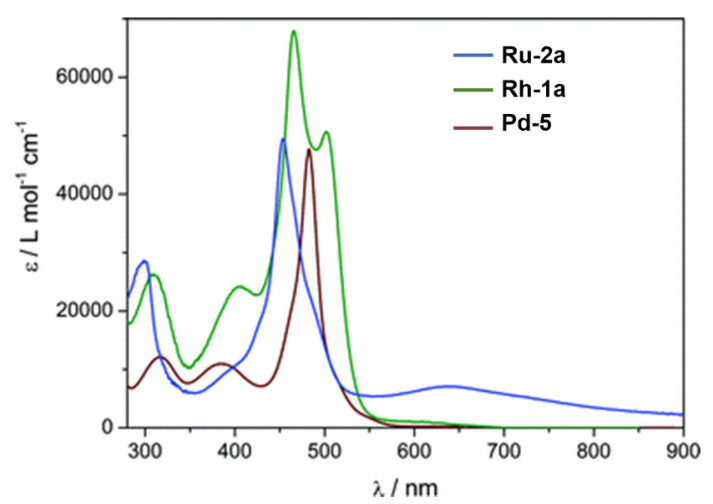
Visible absorption spectra of these complexes in DMSO. Exciton coupling is visible in the absorption spectra of **Rh-1a**. Reproduced with permission from [67]. Copyright © 2010 Royal Society of Chemistry.

**Figure 14 molecules-27-06967-f014:**
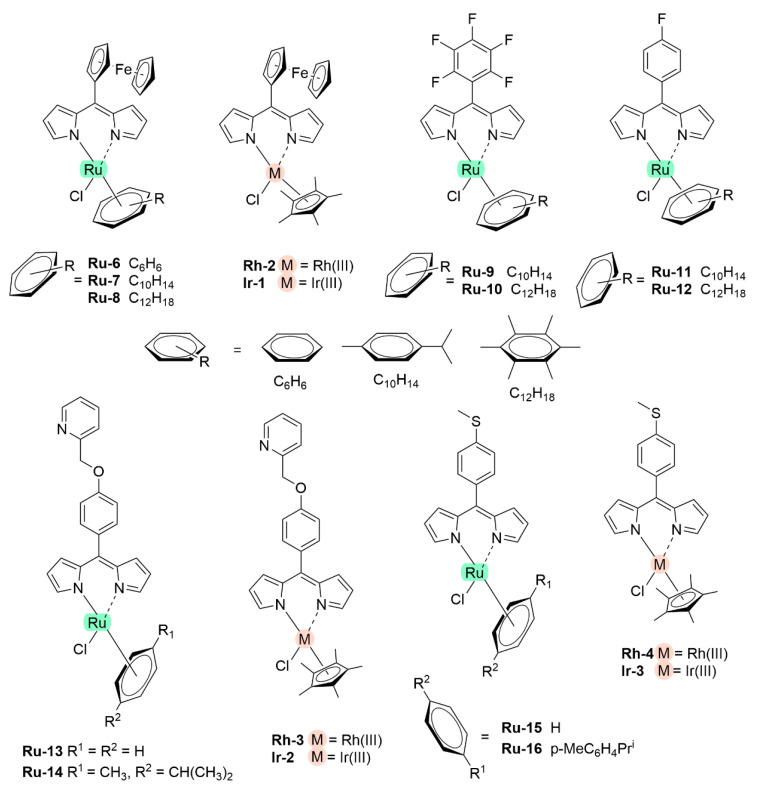
Chemical structures of heteroleptic Ru(II), Rh(III) and Ir(III) dipyrrinato complexes.

**Figure 15 molecules-27-06967-f015:**
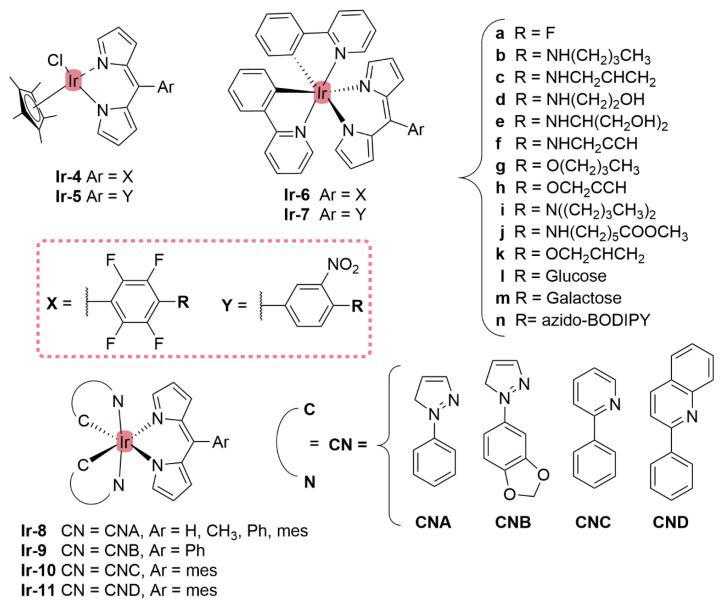
Chemical structures of heteroleptic (dipyrrinato)Ir(III) complexes.

**Figure 16 molecules-27-06967-f016:**
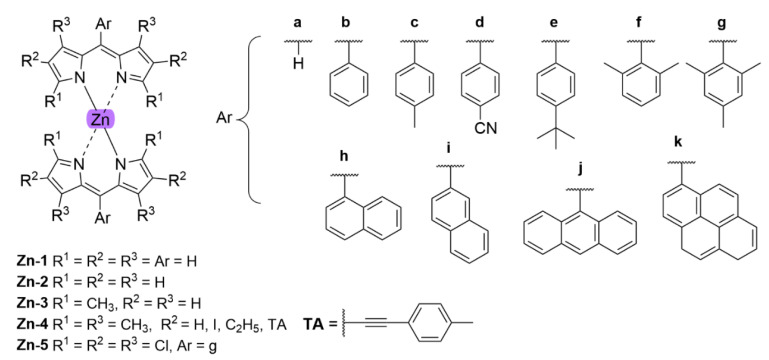
Chemical structures of homoleptic bis(dipyrrinato)Zn(II) complexes (b = Ph, c = C_6_H_4_, d = PhCN, g = Mes).

**Figure 17 molecules-27-06967-f017:**
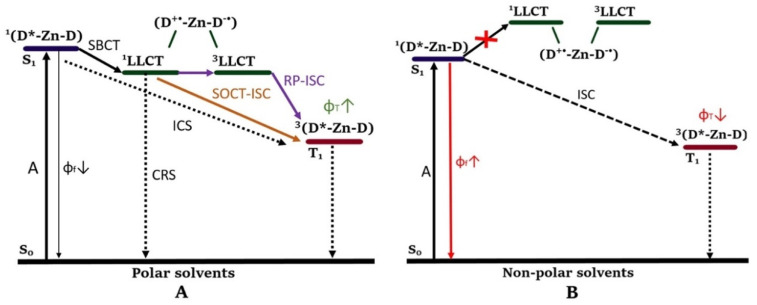
Simplified energy diagram showing the excited-state decay pathways in polar solvents (**A**) and non-polar solvents (**B**). The energy of the (non-emissive) LLCT states is dependent on the solvent. Adapted from [168].

**Figure 18 molecules-27-06967-f018:**
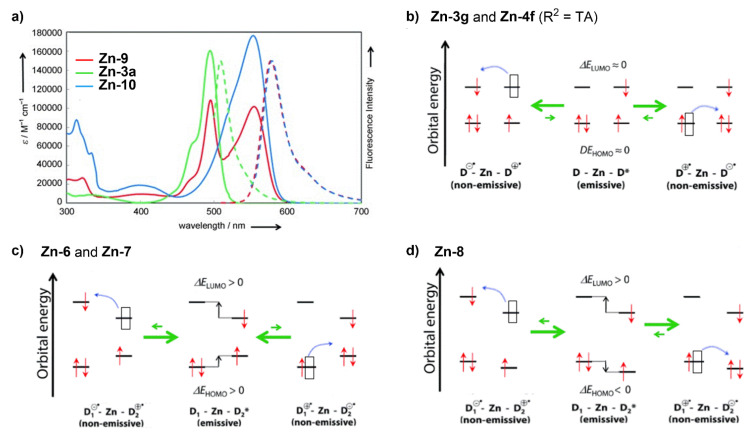
(**a**) Absorption spectra (solid lines) and emission spectra (dotted lines) of the heteroleptic complex **Zn-6** an its homoleptic analogues **Zn-3g** and **Zn-4f** (R^2^ = TA) in toluene. Schematic illustrations of the energy levels and electron transfer processes (blue arrows) in polar solvents for (**b**) homoleptic complexes **Zn-3g** and **Zn-4f** (R^2^ = TA) (**c**) **Zn-6** and **Zn-7**, and (**d**) **Zn-8**. Reproduced with permission from [165]. Copyright © 2012 John Wiley & Sons.

**Figure 19 molecules-27-06967-f019:**
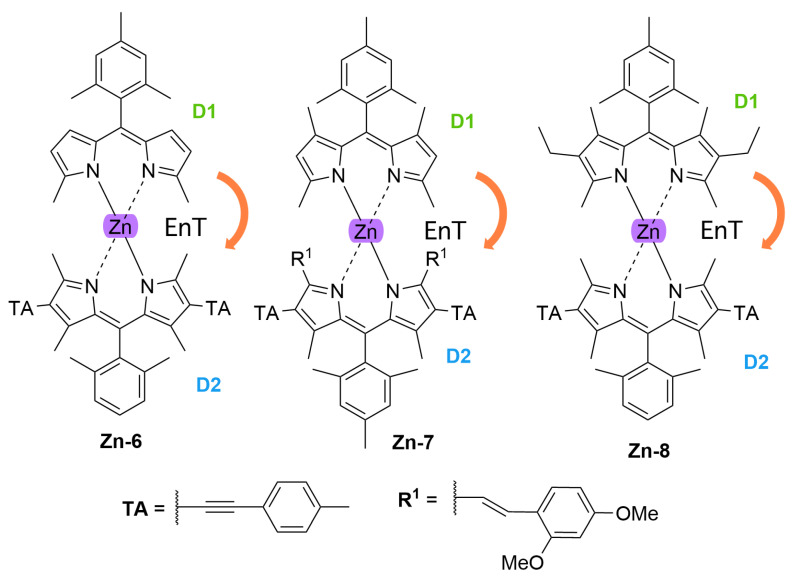
Chemical structures of heteroleptic (dipyrrinato)Zn(II) complexes reported in literature [165,177]. The orange arrow indicates energy transfer (EnT) from the D1 ligand to the D2 ligand.

**Figure 20 molecules-27-06967-f020:**
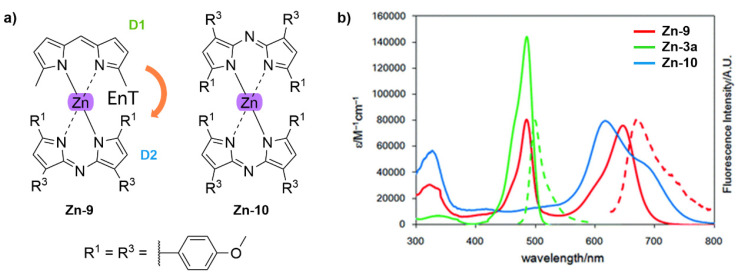
(**a**) Chemical structures and (**b**) UV-Vis absorption (solid line) and fluorescence (dashed line) spectra of **Zn-9, Zn-3a** and **Zn-10**. (**b**) is reproduced with permission from **[120]**. Copyright © 2012 Royal Society of Chemistry.

**Figure 21 molecules-27-06967-f021:**
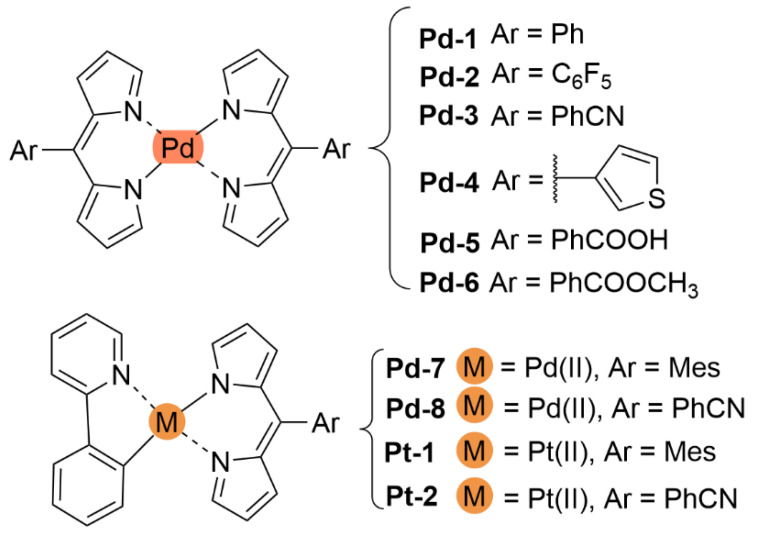
Chemical structures of (dipyrrinato)Pd(II) and -Pt(II) complexes.

**Figure 22 molecules-27-06967-f022:**
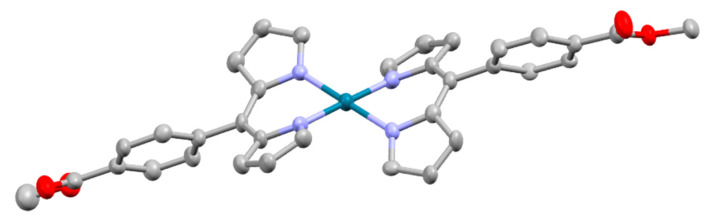
Molecular structure of **Pd-6** as determined by X-ray crystallography by Telfer et al. Hydrogen atoms have been omitted for clarity, thermal ellipsoids show 50% probability [67]. Repulsion between the *α*-hydrogens causes the complex to be in a ’stepped’ square planar configuration. See the main text for more discussion on the properties of this compound. (CCDC number: 737432 [183]).

**Figure 23 molecules-27-06967-f023:**
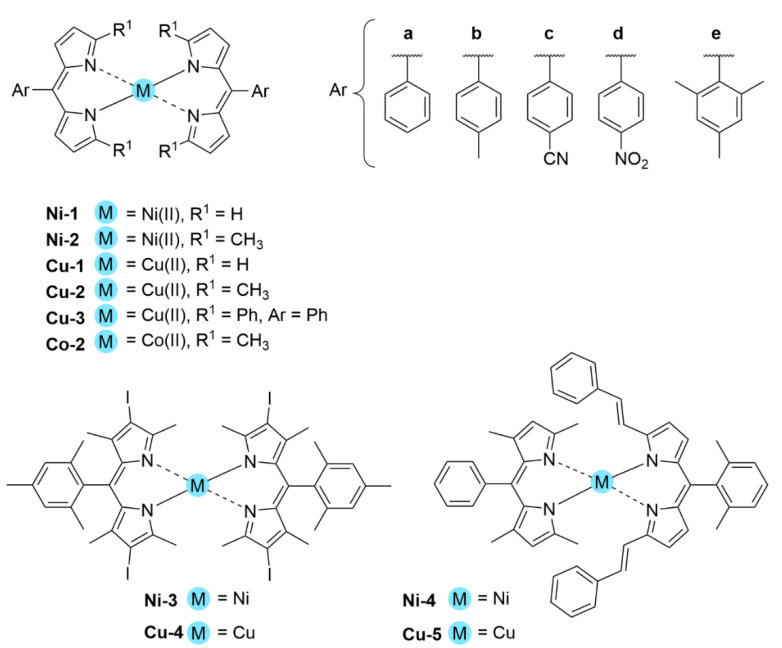
Structures of bis(dipyrrinato)Ni(II) and -Cu(II) and -Co(II) complexes (a = Ph, b = PhMe c = PhCN, d = C_6_H_4_NO_2_, e = Mes).

**Figure 24 molecules-27-06967-f024:**
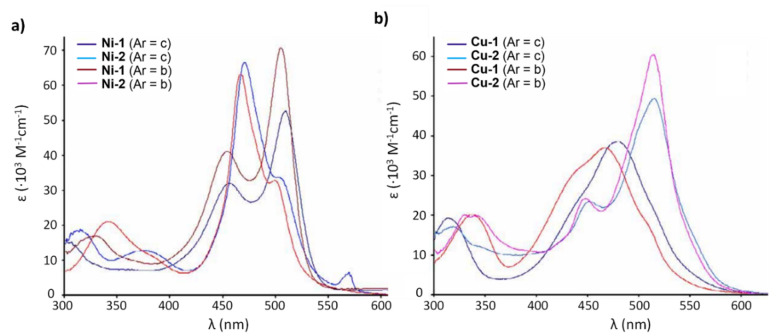
UV-Vis absorption spectra of several (**a**) bis(dipyrrinato)Ni(II) and (**b**) -Cu(II) complexes in DCM. Reproduced with permission from [184]. Copyright © 2009 Royal Society of Chemistry.

**Figure 25 molecules-27-06967-f025:**
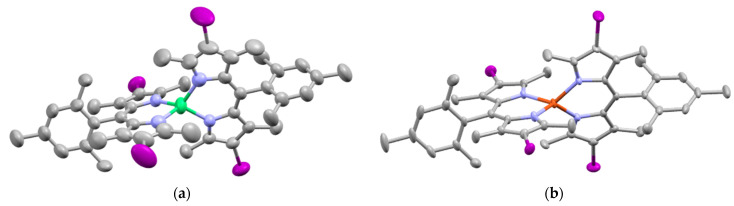
Molecular structures of (**a**) **Ni-3** and (**b**) **Cu-4** in the crystal. Hydrogen atoms have been omitted for clarity, thermal ellipsoids show 50% probability [15]. See the main text for more discussion on the properties of these compounds. (CCDC number: (**a**) 1976221 and (**b**) 1976222 [183]).

**Figure 26 molecules-27-06967-f026:**
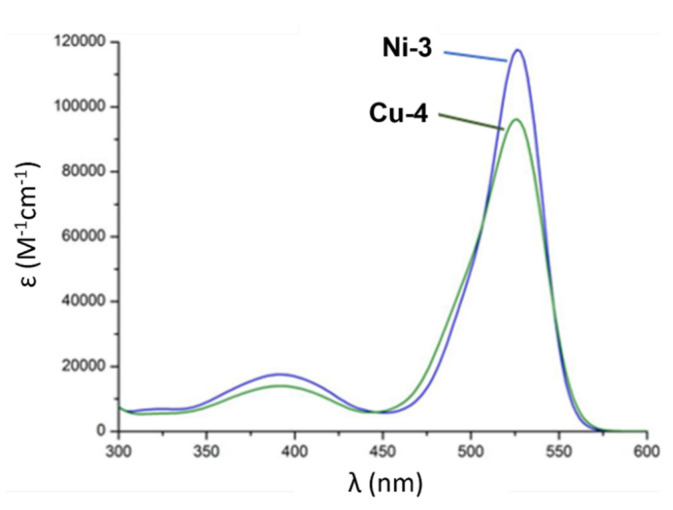
UV-vis absorption spectra of **Ni-3** and **Cu-4** in DCM. Reproduced with permission from [15]. Copyright © 2020 Elsevier B.V.

**Figure 27 molecules-27-06967-f027:**
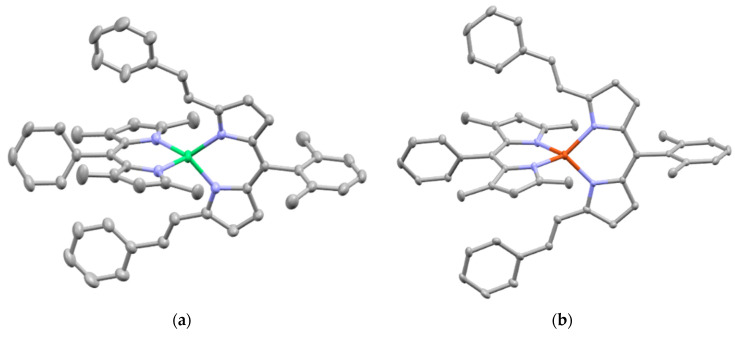
(**a**) View of the molecular structure of **Ni-4** and (**b**) **Cu-5** in the crystal. Hydrogen atoms have been omitted for clarity, thermal ellipsoids show 50% probability [197]. See the main text for more discussion on the properties of these compounds. (CCDC number: 1047912 (**a**) and 1047911 (**b**) [183]).

**Figure 28 molecules-27-06967-f028:**
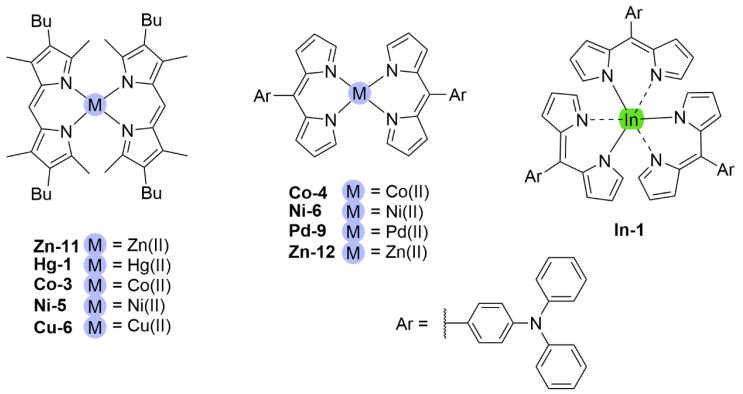
Structures of butyl-substituted complexes **Zn-11**, **Hg-1**, **Co-3**, **Ni-5** and **Cu-6** and triphenylamine substituted complexes **Co-4**, **Ni-6**, **Pd-9**, **Zn-12** and **In-1** [198,201].

**Figure 29 molecules-27-06967-f029:**
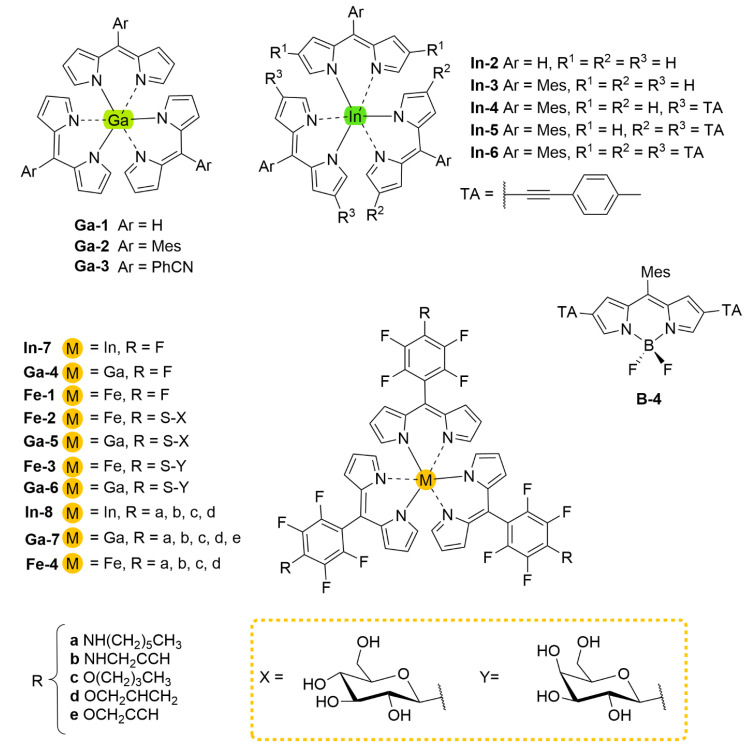
Dipyrrinato-Ga(III) and In(III) as well as Fe(III) complexes reported in literature.

**Figure 30 molecules-27-06967-f030:**
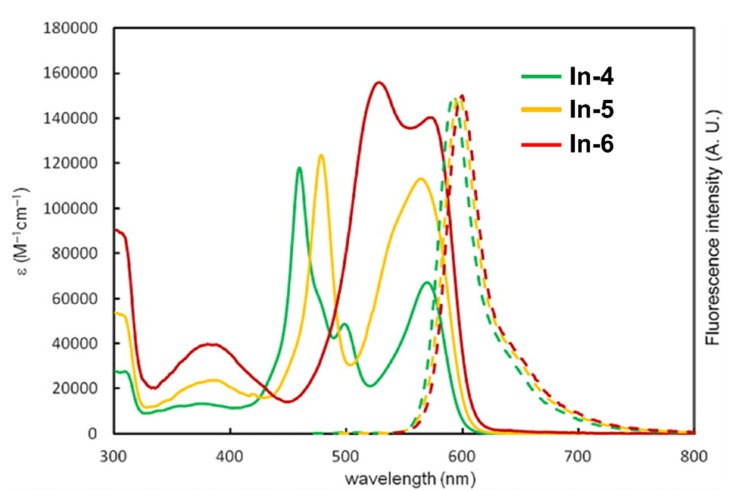
UV-Vis absorption (solid line) and emission (broken line) spectra for **In-4** (green, λ_ex_ = 459 nm), **In-5** (yellow, λ_ex_ = 480 nm) and **In-6** (red, λ_ex_ = 530 nm) in toluene. Reproduced with permission from [203]. Copyright © 2014 American Chemical Society.

**Figure 31 molecules-27-06967-f031:**
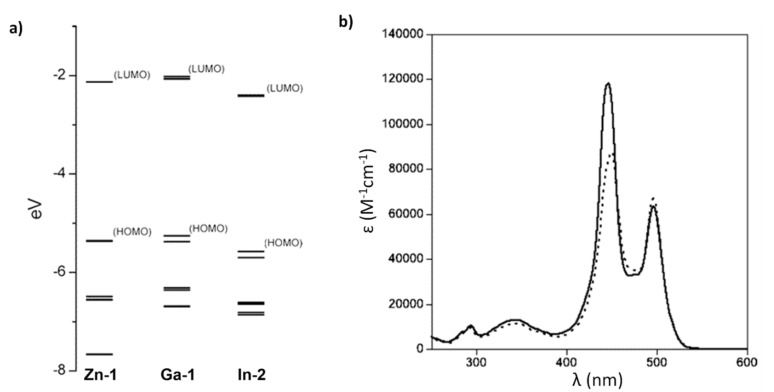
(**a**) Energy level diagrams of **Ga-1**, **In-2** and **Zn-1** obtained with DFT calculations. (**b**) UV-Vis absorption spectra of **Ga-2** (dotted line) and **In-3** (solid line) in hexanes. Reproduced with permission from [3]. Copyright © 2006 American Chemical Society.

**Figure 32 molecules-27-06967-f032:**
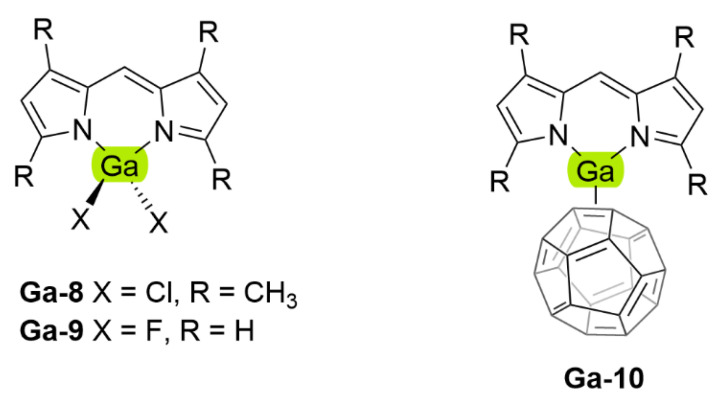
Chemical structures of monomeric Ga(III) complexes (GaDIPYs) reported in literature.

**Figure 33 molecules-27-06967-f033:**
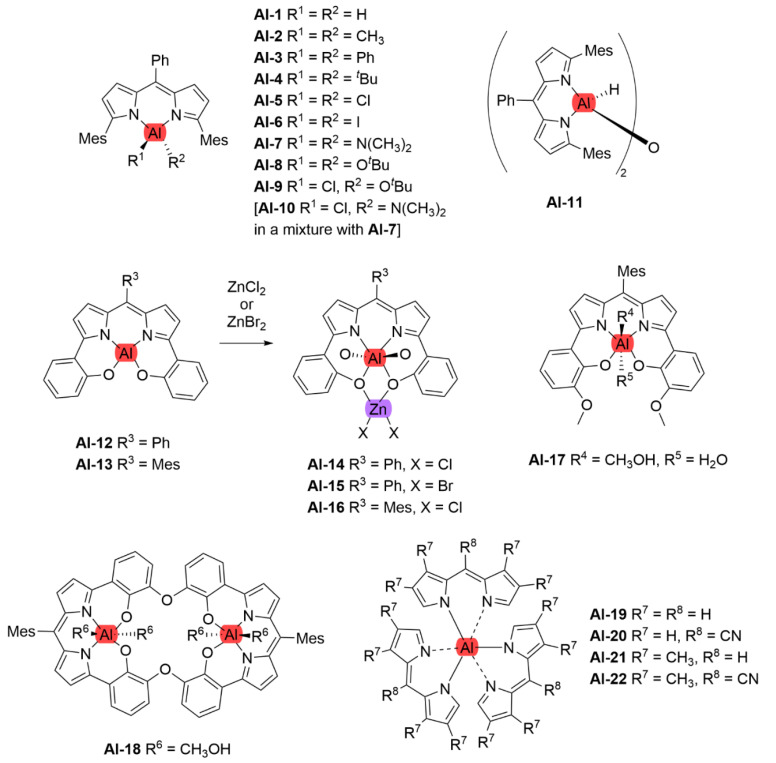
Several (dipyrrinato)Al(III) complexes, such as AlDIPYs **Al-(1–13,17)** and tris(dipyrrinato)Al(III) **Al-(19–22)**, reported in literature [119,204,205].

**Figure 34 molecules-27-06967-f034:**
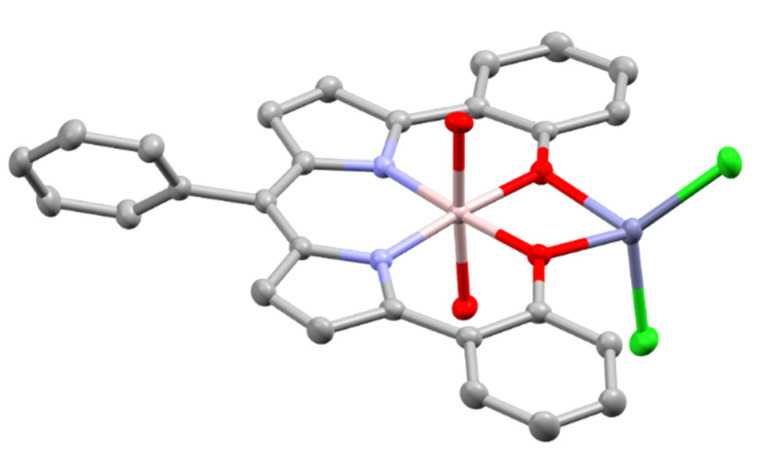
View of the molecular structure of **Al-14** in the crystal, obtained from a 1:1 mixture of aluminum complex **Al-12** with ZnCl_2_. Hydrogen atoms and non-coordinating solvent molecules are omitted for clarity; thermal ellipsoids give 50% probability [119]. The two water molecules in the axial position are represented only by their oxygen atoms. See the main text for more discussion on the properties of this compound. (CCDC number: 708972 [183]).

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
