# Peer review of "Metal Coordination Effects on the Photophysics of Dipyrrinato Photosensitizers"

_molecules, 2022, doi:10.3390/molecules27206967_

Round 1
Reviewer 1 Report
In this review, Paula C. P. Teeuwen et al. summarized the metal coordination effect on the photophysics of metal dipyrrinato complexes. A general overview of the mechanism of photodynamic therapy and the properties of the triplet photosensitizers is given, followed by further details of dipyrrinato complexes described in the literature that show relevance as photosensitizers for PDT. I recommend this review to be published in Molecules after some revisions.
1. Please add the details about the s, d, p blocks in Figure 2. How do they coordinate with dipyrrinato unit?
2. In Figure 3, T1 was mentioned for the mechanism of PDT. Many complexes have no obvious ISC process, such as Fe, Zn, and so on. So how to understand the energy diagram? How to determine the T1 values for most metal complexes?
3. How does the energy level of the metal complex affect the PDT properties?
4. In the review, the author listed many single crystal structures, please give more description of molecular structure and PDT properties.
5. In conclusion, please add more suggestions on the molecular design of metal complexes for PDT application and the outlook for PDT application.
Author Response
We thank referee 1 for the very useful and constructive comments.
- Please add the details about the s, d, p blocks in Figure 2. How do they coordinate with dipyrrinato unit?
We have adapted the Caption of Figure 2, and have added in more information on the coordination of the different elements.
It now reads:
“Octahedral homoleptic tris(dipyrrinato) complexes discussed here, are formed mainly by Al(III), Ga(III), In(III), Fe(III), Co(III) and Rh(III). Tetrahedral or square planar bis(dipyrrinato) complexes are formed mainly by Co(II), Ni(II), Cu(II), Zn(II) and Pd(II). Heteroleptic mono(dipyrrinato) complexes are formed mainly (but not exclusively) with B, Al(III), Ga(III), Ru(II), Rh(III), Pd(II), Re(I), Ir(III) and Pt(II).”
- In Figure 3, T1 was mentioned for the mechanism of PDT. Many complexes have no obvious ISC process, such as Fe, Zn, and so on. So how to understand the energy diagram? How to determine the T1 values for most metal complexes?
We have added text in the Caption of Figure 3, to make it more clear that this is the general mechanism, but that the nature of the states can be changed by the nature of metal ion that complexes with the ligand.
- How does the energy level of the metal complex affect the PDT properties?
We have added text in the Caption of Figure 3, to make it more clear that the energy of the states can also be changed by the nature of metal ion that complexes with the ligand.
- In the review, the author listed many single crystal structures, please give more description of molecular structure and PDT properties.
We now make it more clear that in the main text, the properties and structures of the X-rays are discussed more. We have added this information in every caption of the figures with X-rays. We inserted with every figure of an X-ray:
“See the main text for more discussion on the properties of this compound.”
- In conclusion, please add more suggestions on the molecular design of metal complexes for PDT application and the outlook for PDT application.
For clarity, we have combined our “Conclusions” section with the “Future Prospects” section into one big section that lists all our suggestions together.
Reviewer 2 Report
Figure 3 could be improved by addition of electrons
Figure 6 panel on the right ISC should be subscript
line 585, one lambda should be emission I think so lambda with subscript e would be correct representation
Minor comment: PDT explanation(line 1719-1721) in the conclusion will be better suited if it is done in the introduction, it is more fitting with figure 3.
Author Response
Referee 2
We thank referee 2 for the very useful and constructive comments and for the attention to important details of our work.
Figure 3 could be improved by addition of electrons
Figure 3 is inspired on a figure from our own publication in a recent RSC book (in which the electrons were shown in separate diagrams). Because of the influence of the metal ions in our current work, we have chosen not to show the electrons here, because in principle, the nature of the metal ion can change the nature of the excited states.
Figure 6 panel on the right ISC should be subscript
We have changed the figure 6 accordingly.
line 585, one lambda should be emission I think so lambda with subscript e would be correct representation
We have changed the text on page 585 accordingly.
Minor comment: PDT explanation(line 1719-1721) in the conclusion will be better suited if it is done in the introduction, it is more fitting with figure 3.
We have moved this text to the section of text belonging to Figure 3. It indeed fits better, there.
Reviewer 3 Report
Teeuwen et al. molecules-1956170
In this review article, the authors present a comprehensive review of various dipyrrinato metal complexes and their photophysical properties with a focus on the applications of photodynamic therapy (PDT). This review provides thorough introduction as well as detailed discussions on the relevant literatures on this topic. It is well-written and organized, and it will provide audience with what’s been established and future perspectives on the dipyrrinato metal complexes in the field of PDT.
Some minor comments:
(1) The Jablonski diagram in Figure 3 should show the triplet positioned at a lower energy compared to the singlet. In the current form, they are too close to each other.
(2) In line 113, the authors state that “For both reactions the PS is excited from the ground state (S0) to the excited singlet state (S1).” However, in the diagram, it shows excitation to higher excited-state S2. The authors need to pay attention to it, either change the text or change the diagram.
(3) Why the authors show chlorin in Fig 6 in the BODIPY part? What’s the rational here?
(4) Please make all the color coding for the molecular structures uniform. For example, in Fig 10, the heteroatoms are labelled with different colors.
(5) In Abbreviations and symbols:
a. The authors used ET for energy transfer. ET is commonly referred to electron transfer rather than energy transfer. Maybe EnT is a better abbreviation here.
b. S0 should refer to singlet ground state.
c. T1 should refer to lowest or first triplet state.
(6) Since the review focuses on PDT applications of the dipyrrin metal complexes. Maybe it will be clearer that the title contains “with a focus on PDT applications”.
Summary,
Overall, this manuscript is well-written and thus is recommended for publication in Molecules after minor revisions.
Author Response
Referee 3
We thank referee 3 for the very useful and constructive comments and for the attention to important details of the figures and text of our work.
(1) The Jablonski diagram in Figure 3 should show the triplet positioned at a lower energy compared to the singlet. In the current form, they are too close to each other.
The referee is absolutely right about this. We have changed the figure 3 accordingly.
(2) In line 113, the authors state that “For both reactions the PS is excited from the ground state (S0) to the excited singlet state (S1).” However, in the diagram, it shows excitation to higher excited-state S2. The authors need to pay attention to it, either change the text or change the diagram.
We have changed the text in line 113 accordingly.
(3) Why the authors show chlorin in Fig 6 in the BODIPY part? What’s the rational here?
We have now changed the Title of this section to make more clear that it is about the Heavy Atom Effects, with BODIPY and Chlorins as two examples where these effects can be noticed, which is illustrated by the data in figure 5 and 6.
We have inserted:
“The heavy atom effect in BODIPYs and Chlorins” as title of the section.
and
“In this section we clarify the heavy atom effect, by using BODIPYs and chlorins as examples.”
(4) Please make all the color coding for the molecular structures uniform. For example, in Fig 10, the heteroatoms are labelled with different colors.
We now indicate clearly the colors used in figure 10. They are put into Figure 10 on purpose, to highlight the different coordination modes presented there.
(5) In Abbreviations and symbols:
- The authors used ET for energy transfer. ET is commonly referred to electron transfer rather than energy transfer. Maybe EnT is a better abbreviation here.
- S0 should refer to singlet ground state.
- T1 should refer to lowest or first triplet state.
The referee is absolutely right about this. We have changed the text accordingly.
We have therefore also updated figure 19 and Figure 20 by inserting “EnT”.
(6) Since the review focuses on PDT applications of the dipyrrin metal complexes. Maybe it will be clearer that the title contains “with a focus on PDT applications”.
We thank the referee for this suggestion, but we do feel that our work is much more about the sensitizers than about PDT. It is more about the photophysics than about actual biological effects, and thus about the more general applications as photosensitizers. Therefore, we have decided that we will keep the title in the current form.